# Differential functions of the dorsal and intermediate regions of the hippocampus for optimal goal-directed navigation in VR space

**Hyeri Hwang[1], Seung-Woo Jin[2], Inah Lee[1]***

[1]Department of Brain and Cognitive Sciences, Seoul National University, Seoul, Republic of Korea; [2]Department of Psychiatry and Behavioral Sciences, University of Washington, Seattle, United States

*****For correspondence:
inahlee@snu.ac.kr

**Abstract** Goal-directed navigation requires the hippocampus to process spatial information in a value-dependent manner, but its underlying mechanism needs to be better understood. Here, we investigated whether the dorsal (dHP) and intermediate (iHP) regions of the hippocampus differentially function in processing place and its associated value information. Rats were trained in a place-preference task involving reward zones with different values in a visually rich virtual reality environment where two-dimensional navigation was possible. Rats learned to use distal visual scenes effectively to navigate to the reward zone associated with a higher reward. Inactivation of both dHP and iHP with muscimol altered the efficiency and precision of wayfinding behavior, but iHP inactivation induced more severe damage, including impaired place preference. Our findings suggest that the iHP is more critical for value-dependent navigation toward higher-value goal locations.

## eLife assessment

The authors report **solid** evidence for a **valuable** set of findings in rats performing a new virtual place-preference task. Temporary pharmacological inhibition targeting the dorsal or intermediate hippocampus disrupted navigation to a goal location in the task, and functional inhibition of the intermediate hippocampus was more detrimental than functional inhibition of the dorsal hippocampus. The work provides novel insights into functional differentiation along the dorsal-ventral axis of the hippocampus.

## Introduction

It has long been suggested that the hippocampus is the neural substrate of a 'cognitive map' – a map-like representation of the spatial environment that allows flexible spatial navigation (***O'Keefe and Nadel, 1978***). The cognitive map is also needed for remembering important events in space. Animals in the natural environment often navigate to achieve goals, such as finding food or avoiding predators, and this goal-directed navigation involves remembering places and their associated values. It has been reported that the receptive fields of place cells in the hippocampus tend to accumulate near a goal location or shift toward it (***Hollup et al., 2001***; ***Kennedy and Shapiro, 2009***; ***Dupret et al., 2010***). One could argue that the hippocampus must process task- or goal-relevant information, including the value of a place, to achieve the goal. However, the specific hippocampal processes involved in integrating the two types of representation – place and value – toward goal-oriented behavior are still largely unknown.

Such integration may occur along the dorsoventral axis of the hippocampus. Previous anatomical studies (*Krettek and Price, 1977*; *Swanson et al., 1978*; *Pikkarainen et al., 1999*; *Tao et al., 2021*) suggest that the hippocampus can be divided along its dorsoventral axis into dorsal (dHP), intermediate (iHP), and ventral (vHP) hippocampal subregions based on different anatomical characteristics. The dHP is connected with brain regions that process visuospatial information, including the retrosplenial cortex and the caudomedial entorhinal cortex (*Van Groen and Wyss, 2003*; *Dolorfo and Amaral, 1998*); it also communicates with the iHP via bidirectional extrinsic connections but exhibits limited connections with the vHP (*Tao et al., 2021*; *Swanson et al., 1978*). The iHP receives heavy projections from valence-related areas, such as the amygdala and ventral tegmental area (VTA) – subcortical inputs that are less prominent in the dHP (*Pikkarainen et al., 1999*; *Felix-Ortiz and Tye, 2014*; *Gasbarri et al., 1994*). The vHP also has connections with the iHP and value-representing areas such as the amygdala, but it does not project heavily to the dHP (*Tao et al., 2021*; *Swanson et al., 1978*; *Pikkarainen et al., 1999*; *Krettek and Price, 1977*). Notably, compared with the dHP, the iHP and vHP have much heavier connections with the medial prefrontal cortex (mPFC), contributing to goal-directed action control (*Hoover and Vertes, 2007*; *Liu and Carter, 2018*). Additionally, the three subregions along the dorsoventral axis display different gene expression patterns that corroborate the anatomical delineations (*Dong et al., 2009*; *Bienkowski et al., 2018*). Overall, the iHP subregion of the hippocampus appears to be ideally suited to integrating information from the dHP and vHP.

Surprisingly, beyond the recognition of anatomical divisions, the available literature on the functional differentiation of subregions along the dorsoventral axis of the hippocampus, particularly in the context of value representation, is somewhat inconsistent. Specifically, there is physiological evidence that the size of a place field becomes larger as recordings of place cells move from the dHP to the vHP (*Jung et al., 1994*; *Maurer et al., 2005*; *Kjelstrup et al., 2008*; *Royer et al., 2010*). Thus, it has been thought that the dHP is more specialized for fine-grained spatial representation than the iHP and vHP. However, when it comes to the neural representation of value information of a place, results are mixed. Several studies have reported that place fields recorded in the dHP respond to internal states and motivational significance based on their accumulation near behaviorally significant locations (e.g., reward locations or escape platforms; *Hollup et al., 2001*; *Kennedy and Shapiro, 2009*; *Dupret et al., 2010*), or to the reward per se (*Gauthier and Tank, 2018*). In contrast, others have reported that dHP place cells do not alter their activity according to a change in reward or reward location and thus do not represent value information (*Duvelle et al., 2019*; *Jin and Lee, 2021*; *Speakman and O'Keefe, 1990*).

Furthermore, although the iHP and vHP have mainly been studied in the context of fear and anxiety, several studies have also reported spatial representation and value-related signals in these subregions. Specifically, prior studies reported that rats with a dysfunctional dHP retained normal goal-directed, target-searching behavior if the iHP and vHP were intact (*Moser et al., 1995*; *de Hoz et al., 2003*). Moreover, lesions in the iHP have been shown to impair rapid place learning in the water maze task (*Bast et al., 2009*). Our laboratory also reported that place cells in the iHP, but not the dHP, instantly respond to a change in spatial value and overrepresent high-value locations (*Jin and Lee, 2021*).

Based on existing experimental evidence, we hypothesize that the iHP is the primary locus for associating spatial representation with value information, distinguishing it from the dHP and vHP. In the current study, we investigated the differential functions of the dHP and iHP in goal-directed spatial navigation by monitoring behavioral changes after pharmacological inactivation of either of the two regions as rats performed a place-preference task in a two-dimensional (2D) virtual reality (VR) environment. In this experimental paradigm, rats learned to navigate toward one of two hidden goal locations associated with different reward amounts. Whereas inactivation of the dHP mainly affected the precision of wayfinding, iHP inactivation impaired value-dependent navigation more severely by affecting place preference.

## Results

### Well-trained rats align themselves toward the high-value zone before departure in the place-preference task

We established a VR version of a place-preference task (*Figure 1A*) in which rats could navigate a 2D environment by rolling a spherical treadmill with their body locations fixed, allowing them to run

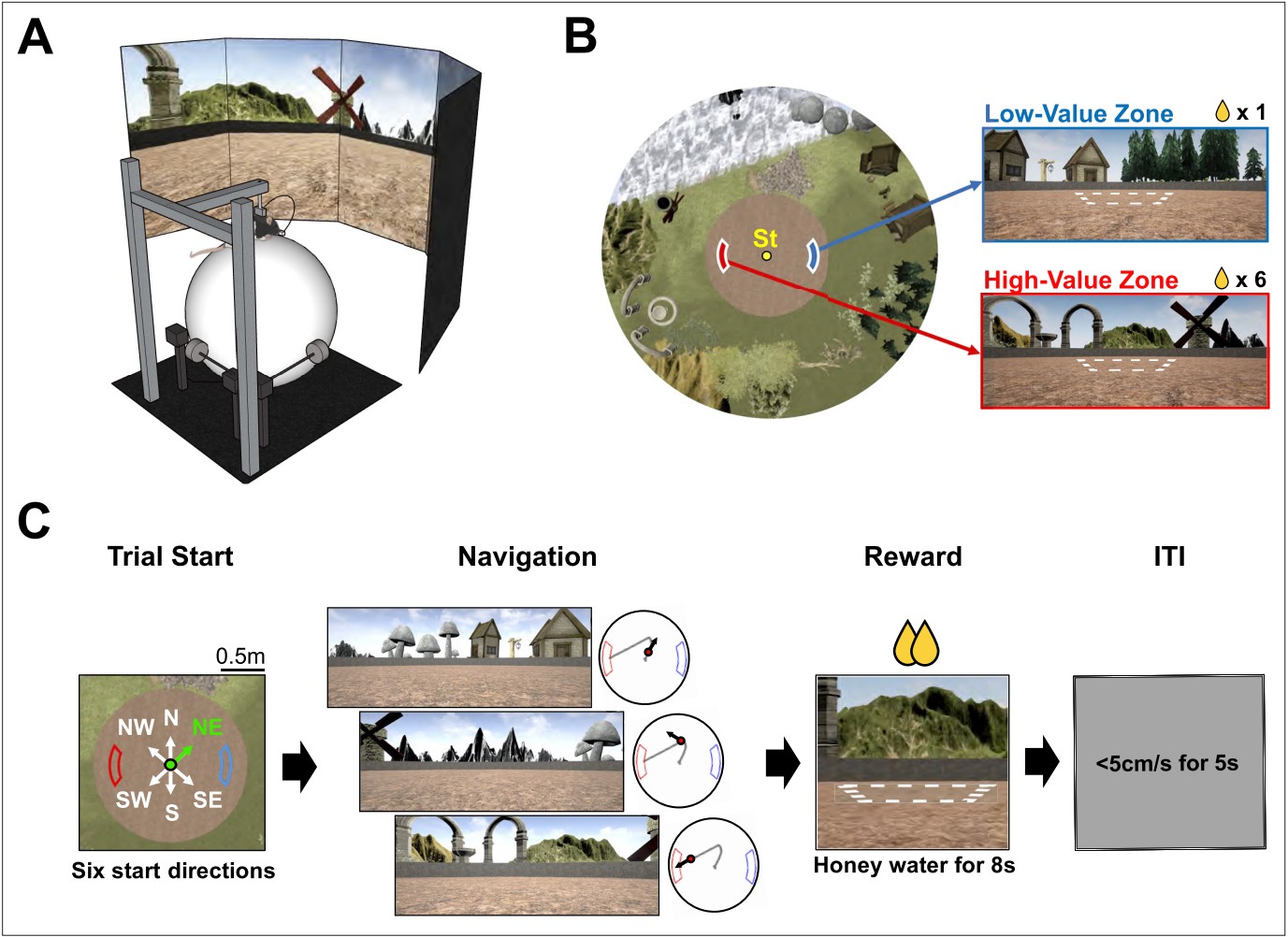

**Figure 1.** Place-preference task in a 2D virtual reality (VR) environment. (**A**) 2D VR setup. (**B**) Bird's-eye view of the virtual environment. Various landmarks surrounded a circular arena, and a fixed start location ('St') was at the center. Reward zones are illustrated with white dashed lines for visualization purposes. (**C**) Place-preference task paradigm. A trial started with one of six pseudorandomly chosen start directions ('Trial Start'). In this example, the rat started the trial facing the northeast (NE) direction, highlighted in green. Subsequent navigation is illustrated here with the associated scene ('Navigation'). A dot on the gray trajectory indicates the rat's current location, and the black arrow describes the head direction. When the rat arrived at a reward zone, honey water was delivered within 8 s, with the visual scene frozen ('Reward'). Finally, a gray screen appeared, denoting an inter-trial interval; if the rat remained still (<5 cm/s) for 5 s, the subsequent trial began ('ITI').

at the apex of the treadmill. Body-fixed rats (n = 8) were trained to explore a virtual circular arena surrounded by multiple distal visual landmarks (houses, rocks, mountains, and trees) (*Figure 1B*). Rats were always started at the center of the arena, and the arena contained two unmarked reward zones – a high-value zone and a low-value zone – each associated with different amounts of honey water (6:1 ratio between high- and low-value zones). A trial started with the rat facing one of six start directions – north (N), northeast (NE), southeast (SE), south (S), southwest (SW), and northwest (NW) – determined pseudorandomly to guarantee equal numbers of trials in all directions. In the example trial shown in *Figure 1C*, the rat was heading in the NE direction at the start location ('Trial Start'), then turned to the left side to run toward the W goal zone ('Navigation'). Once the rat arrived at one of the reward zones, the synchronization between the spherical treadmill movement and the virtual environment stopped, and multiple drops of honey water were delivered via the licking port ('Reward'). Then, during an inter-trial interval (ITI), the LCD screens turned gray, and the rat was required to remain still for 5 s to initiate the subsequent trial. The pre-surgical training session consisted of 60 trials, which were reduced to 40 during post-surgical training.

Pre-surgical training began after shaping in the VR environment. On average, it took 13 days for rats to reach pre-surgical criteria, namely, to complete 60 trials and visit the high-value goal

zone in more than 75% of completed trials (see 'Materials and methods' for the detailed performance criteria). Well-trained rats exhibited two common behaviors during the pre-surgical training. First, although it was not required in the task, they learned to rotate the spherical treadmill counterclockwise to move around in the virtual environment (presumably to perform energy-efficient navigation). To rule out the potential effect of hardware bias or any particular aspect of peripheral landscape to make rats turn only to one side, we measured the direction of the first body-turn in each trial on the last day of shaping and the first day of the main task (i.e., before rats learned the reward zones). There was no significant difference between the clockwise and counterclockwise turns (p=0.46 for shaping, p=0.76 for the main task; Wilcoxon signed-rank test), indicating that the stereotypical pattern of counterclockwise body-turn appeared only after the rats learned the reward locations.

Second, once a trial started, the animal rotated the treadmill immediately to align its starting direction with the visual scene associated with the high-value reward zone. After setting the starting direction, the rat started to run on the spherical treadmill, moving the treadmill forward to navigate directly toward the reward zone. All eight rats displayed this strategy during the later learning phase but not during the earlier learning stage, suggesting that the start-scene–alignment strategy was learned during training. Because the initial rotational scene alignment before departure was an essential component of the task and this behavior was not readily detectable with position-based analysis, we based most of our behavioral analysis on the directional information defined by the allocentric reference frame of the virtual environment (*Figure 2A*). Because we did not measure the rat's head direction in the current study, the allocentric directional information represented the angular position of a particular scene in the virtual environment displayed in the center of the screen.

To establish the direction in which the rat departed the starting point at the center after scene alignment, we first defined a *departure circle* – a virtual circle (~20cm in diameter) in the VR environment at the center of the arena (*Figure 2A*). In the example trial shown in *Figure 2A*, the rat faced the NE direction (315°) at the trial start but immediately turned its body to the NW direction upon starting and ran straight toward the high-value zone after that. Since the initial scene rotations at the start point cannot be visualized in the position-based graph, we made a *scene rotation plot* that visualizes the rotational movement traces in the virtual environment. The scene rotation plot covers the period from the start of the trial to when the rat leaves the departure circle (*Figure 2B*).

On the first day of training for the task ('Novice' stage), the rat produced almost no rotation of the VR environment until he exited the departure circle, indicating that the animal ran straight in the initially set start direction without adjusting the scene orientation. As a result, rats missed the target reward zones in most trials (*Figure 2B and C*). However, by the last day of training ('Expert' stage), there were noticeable rotational shifts in all directional traces (i.e., counterclockwise rotations) that converged on the high-value reward zone (*Figure 2B and D*). This was the case for all trials except those in which the initial start direction almost matched the orientation of the high-value reward zone (i.e., 225° or NW). Furthermore, the average travel distance and latency for each start direction declined from the novice to the expert stage, suggesting that the rats navigated more efficiently toward reward zones in the later stage of learning by pre-adjusting their starting scene direction at the trial start (*Figure 2C and D*).

Overall, the marked differences in orienting behaviors between early and late learning stages suggest that rats could discriminate the high-value reward zone from the low-value zone in our VR environment and show that they preferred visiting the high-value reward zone over the low-value zone. It also indicates that rats could explore the VR environment using allocentric visual cues to find the critical scenes associated with the high-value zone before leaving the starting point (i.e., departure circle).

## Departing orientation and perimeter-crossing direction provide a measure of navigational efficiency and precision, respectively

To analyze behavioral changes during learning in more detail, we analyzed various learning-related parameters at different stages of pre-surgical training. For this, we focused on days in which rats visited the high-value zone on more than 75% of trials for two consecutive days – the performance criterion for completion of pre-surgical training. These two consecutive days (post-learning days) were grouped and averaged for each rat as the post-learning group ('POST' in *Figure 3Aii*) and compared

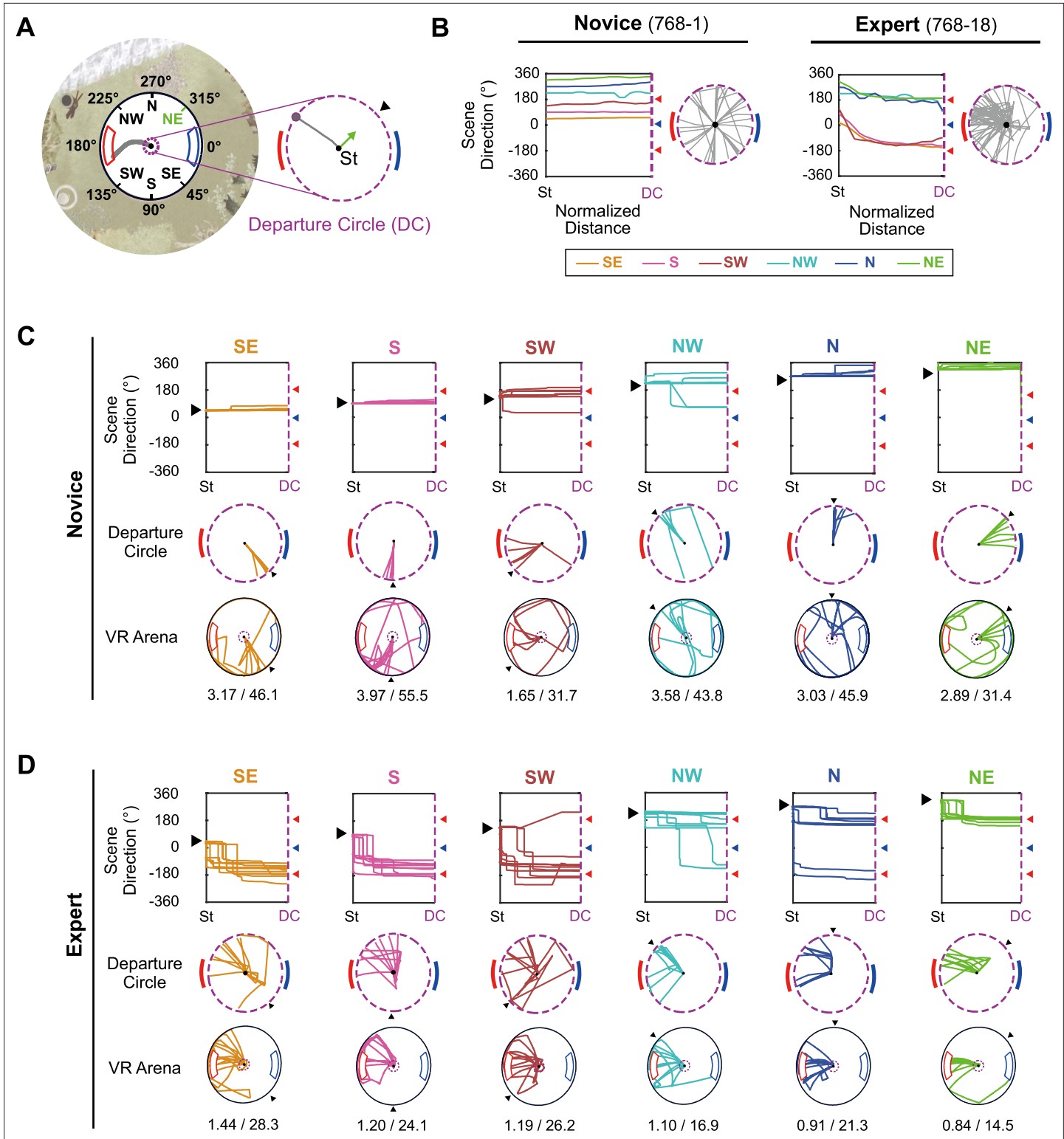

**Figure 2.** Common body-turning behavior of rats after learning. (**A**) The reference frame of the virtual environment. The six start directions are illustrated with the red high-value zone (180°) and blue low-value zone (0°). On the right, the departure circle (DC) is denoted with a purple dashed line, and the start direction is marked with a black arrowhead and a green arrow. (**B**) Overall changes in scene direction over the normalized distance between the start location and the DC (left). Each colored line indicates the median change of scene direction in trials with each start location, and red and blue arrowheads mark high- and low-value zone centers, respectively. The 0°-to-360° range was repeated in the ordinate of the plot to capture rotational movements in opposite directions (positive and negative directions for clockwise and counterclockwise rotations, respectively). The gray

*Figure 2 continued on next page*

*Figure 2 continued*

lines on the right show the rat's trajectory within the DC. These examples were excerpted from the first and last days of pre-training of a single rat. The numbers after 'Novice' and 'Expert' indicate the rat and session number of the example. (C) Individual examples of scene directions and trajectories in the novice session. Scene direction change for each direction is drawn separately (top) for individual trials. The black arrowhead indicates that specific start direction. Trajectories within the DC (middle) and the whole arena (bottom) are also illustrated according to the indicated color code. Mean travel distance in meters and latency in seconds are shown below the virtual reality (VR) arena trajectory. (D) Same as (C), but for the expert session.

with the two consecutive days immediately preceding the post-learning days (pre-learning days; 'PRE' in *Figure 3Aii*).

We first measured the departing direction when crossing the departure circle (departing direction [DD]; *Figure 3Ai*). As indicated in *Figure 2*, well-trained rats rotated the VR environment to place the target VR scene (i.e., high-value reward zone scene) ahead before departure. Therefore, alignment of the DD with the high-value zone at the beginning of navigation indicated that the rat remembered the scenes associated with the high-value zone. For example, the distribution of pre-learning days DDs was widely distributed without any directional bias; as such, its mean vector was small (*Figure 3Aii*). On the other hand, the DDs of post-learning sessions mostly converged on the direction aligned with the high-value zone, resulting in a larger mean vector length compared with that in the pre-learning session. The distributions of averaged DDs for all rats significantly differed between 'PRE' and 'POST' (p<0.001, Kuiper's test), verifying that DD is a valid index of the acquisition of the high-value zone and efficient navigation. To investigate how accurately rats oriented themselves directly to the high-value zone before leaving the start point, we also calculated the average deviation angle of DDs

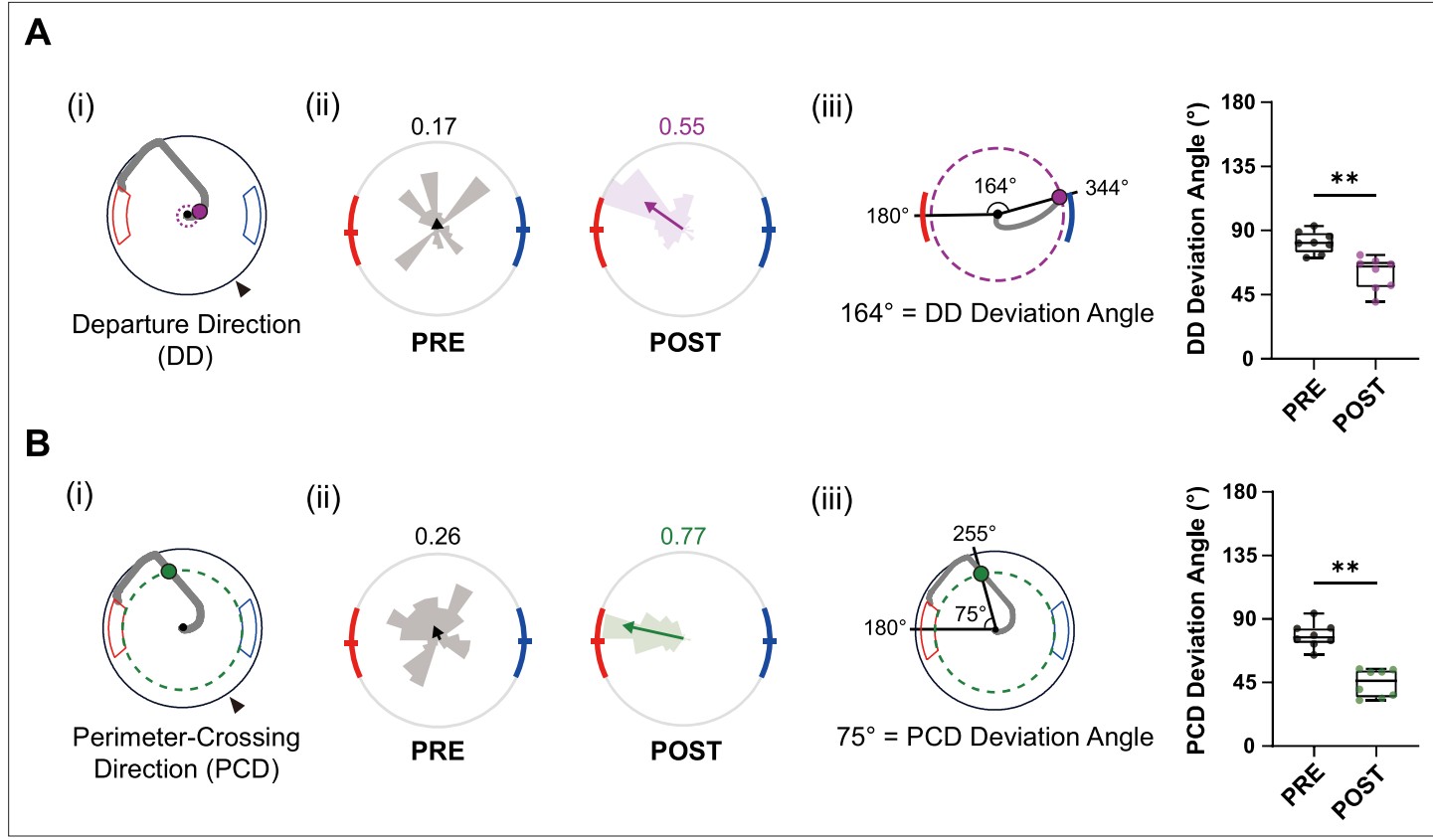

**Figure 3.** Learning index for efficient navigation during pre-surgical training. (A) Changes in departing direction (DD) with learning. (i) Schematic of DD (purple dot), with the departure circle shown as a dashed line. (ii) Distribution of DDs in pre- and post-learning sessions from all rats (rose plots). Gray denotes the pre-learning session, whereas purple indicates the post-learning session. Mean vectors are illustrated as arrows with the same color scheme, and their lengths are indicated at the upper right side of the plot. (iii) Schematic of the DD-deviation angle (angle between the high-value zone center and the DD) and comparisons of DD-deviation angles between pre- and post-learning sessions. Each dot represents data from one rat (n=8). (B) Same as (A), but for perimeter-crossing direction (PCD; green dot). The perimeter is drawn as a green dashed circle. Data are shown as box plots (**p<0.01, Wilcoxon signed-rank test), and the significance level was set at α = 0.05.

(DD-deviation) – the angle between the DD and the high-value zone (180°, measured at the center of the zone) – for each rat (*Figure 3Aiii*). A comparison between pre- and post-learning sessions showed that the DD-deviation significantly declined after learning. This implies that well-trained rats aligned their bodies more efficiently to directly navigate to the high-value zone (p<0.01, Wilcoxon signed-rank test).

Rats adjusted their navigational routes further, even after exiting the departure circle, to navigate more accurately and straight to the goal, avoiding the wall surrounding the arena. Such fine spatial tuning (i.e., navigation precision), measured as the decrease in DD-deviation, only appeared after the rats learned the high-value reward location. To quantify navigation precision, we measured the perimeter-crossing direction (PCD; *Figure 3Bi*), defined as the angle at which the rat first touched the unmarked circular boundary along the arena's perimeter, which shares the inner boundaries of the reward zones (green dashed lines in *Figure 3Bi*). During pre-learning, the PCD was randomly distributed along the perimeter ('PRE' in *Figure 3Bii*). On the other hand, in most post-learning stage trials, rats crossed the unmarked peripheral boundaries only in the vicinity of the high-value zone ('POST' in *Figure 3Bii*). Since rats usually turned counterclockwise during navigation, the convergence of crossings near the northern edge of the high-value zone indicates that they took a shortcut – the most efficient route – to enter the goal zone. The PCD distributions were significantly different between pre- and post-learning stages (p<0.001, Kuiper's test) (*Figure 3Bii*). The deviation angle between the PCD and the high-value zone center also significantly decreased with learning (*Figure 3Biii*), indicating that the navigation of rats to the goal became more accurate.

Additionally, to investigate whether the rats used a certain landmark as a beacon to find the reward zones, we conducted the landmark omission test as a part of control experiments. Here, one of the landmarks was omitted, and the landmark to be made disappear was pseudorandomly manipulated on a trial-by-trial basis. The omission of one landmark, regardless of its identity, did not cause a

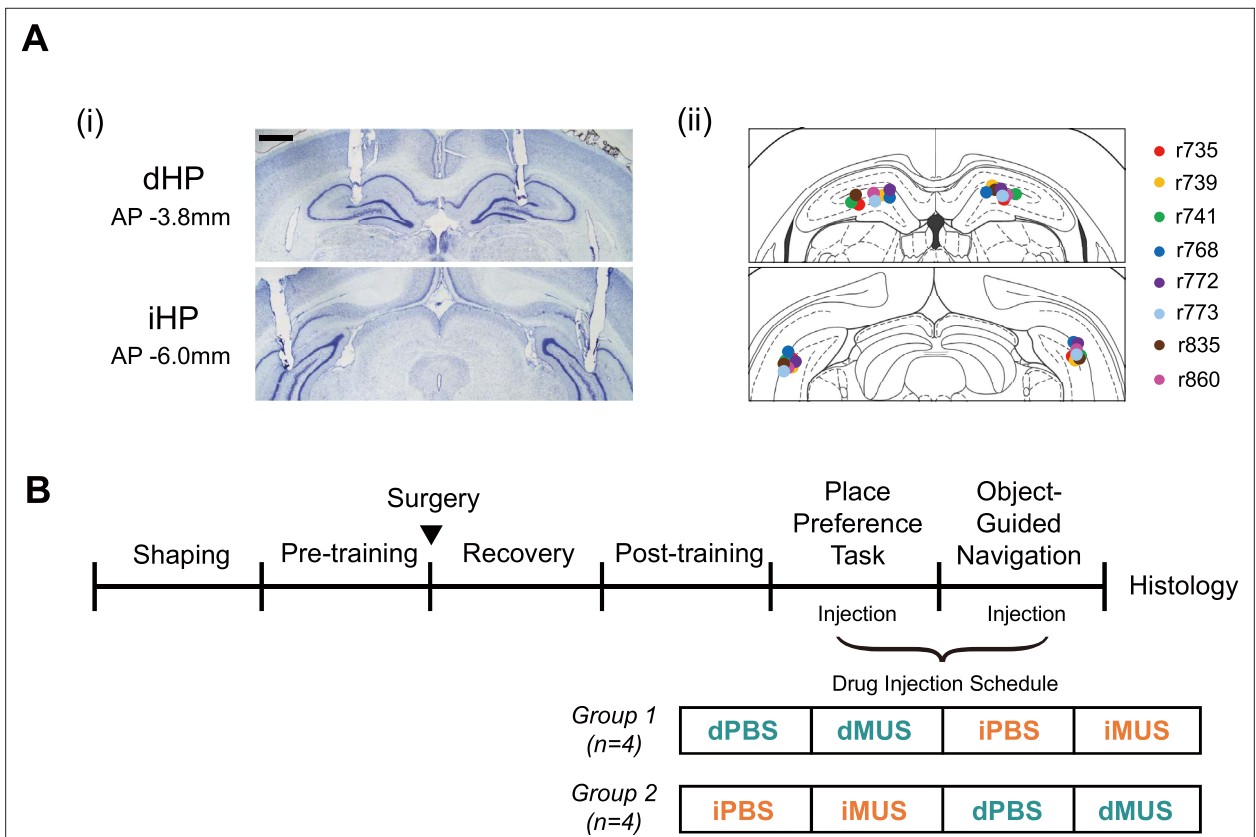

**Figure 4.** Cannula implantation locations and schedules for training and drug injection. (**A**) Cannula positions marked. The scale bar at the upper left indicates 1mm. (**i**) Example of bilaterally implanted cannula tracks in Nissl-stained sections in the dorsal hippocampal (dHP) and intermediate hippocampal (iHP). (**ii**) Tip locations illustrated in the atlas, with different colors for individual rats (n = 8). (**B**) Training schedule. Rats were divided into two groups (n = 4/group) to counterbalance the injection order for the main task and probe test.

specific behavioral change in finding the reward zones, suggesting that the rats were not relying on a single visual landmark when finding the reward zones. The result can be reported anecdotally only because of an insufficient sample size (n = 3), not permitting any meaningful statistical testing.

## Navigation is impaired by inactivation of either the dHP or iHP, but only iHP inactivation affects place-preference behavior

To dissociate the roles of the dHP and iHP, we inactivated either the dHP or iHP in an individual animal using muscimol (MUS), a GABA-A receptor agonist, before the rat performed the place-preference task. To allow within-subject comparisons in performance, we bilaterally implanted two pairs of cannulas – one targeting the dHP and the other targeting the iHP – in the same rat after it successfully reached pre-surgical training criteria (*Figure 4A*). After 1 week of recovery from surgery, rats were retrained to regain a level of performance similar to that in the pre-surgical training period (*Figure 4B*), after which the drug injection schedule was started.

We divided rats into two drug injection groups (n = 4 rats/group) to counterbalance the injection order between the dHP and iHP. Rats in one group received drug infusion into the dHP first, whereas rats in the other group were injected into the iHP first. For all rats, phosphate-buffered saline (PBS) was initially injected in both regions as a vehicle control. For analytical purposes, we first ensured no statistical difference in performance between the two PBS sessions (dPBS and iPBS; see below) and then averaged them into a single PBS session to increase statistical power. During the PBS session, rats tended to take the most efficient path to the high-value zone, as they had done during pre-surgical training (*Figure 5A*). They aligned the VR scene at the start with the high-value zone for all start directions and then ran directly toward the goal zone. Notably, once the start scene alignment was complete, rats usually moved quickly and straight without slowing in the middle. Also, their navigation paths led them directly toward the center of the goal zone. During subsequent dHP-inactivation sessions, rats appeared less accurate, bumping into the arena wall in many trials (dMUS in *Figure 5A*), but most of these wall bumps occurred in the vicinity of the high-value zone, and rats quickly compensated for their error by turning their bodies to target the reward zone correctly after wall bumping. In contrast, in iHP-inactivation sessions, the trajectories were largely disorganized, and the wall-bumping locations were no longer limited to the vicinity of the high-value zone. In some trials, rats moved largely randomly (as shown in 860-17-24 in *Figure 5A*) and appeared to visit the low-value zone significantly more than during PBS or dMUS sessions.

To quantitatively analyze these observations, we compared the proportions of visits to the high-value zone among drug conditions (*Figure 5B*), finding a significant difference in the percentage of correct target visits among drug conditions ($F_{(2,14)}$ = 10.56, p<0.01, one-way repeated-measures ANOVA; p=0.25 for dPBS vs. iPBS, Wilcoxon signed-rank test). A post hoc analysis revealed a significant decrease in the iMUS session compared to the PBS session (p<0.05, Bonferroni-corrected post hoc test). In contrast, no significant differences were found in other conditions, although there was a decreasing trend in the iMUS compared to PBS (p=0.2 for PBS vs. dMUS; p=0.1 for dMUS vs. iMUS). These results indicate that dHP-inactivated rats showed a strong preference for the high-value zone, as they did in control sessions, but that the performance of iHP-inactivated rats was impaired in our place-preference task, as reflected in their significantly more frequent visits to the low-value zone compared with controls.

It is unlikely that these differences stemmed from generic sensorimotor impairment as a result of MUS infusion because running speed remained unchanged across drug conditions ($F_{(2,14)}$ = 0.99, p=0.37, one-way repeated-measures ANOVA; p=0.95 for dPBS vs. iPBS, Wilcoxon signed-rank test) (*Figure 5C*). Furthermore, rats remained motivated throughout the testing session, as evidenced by the absence of a significant difference in the number of trials across drug groups ($F_{(1.16, 8.13)}$=1.34, p=0.29, one-way repeated-measures ANOVA with Greenhouse–Geisser correction; p=1.0 for dPBS vs. iPBS, Wilcoxon signed-rank test; data not shown), although there was an increase in the session duration ($F_{(2,14)}$ = 6.46, p<0.05, one-way repeated-measures ANOVA; p=0.27 for dPBS vs. iPBS, Wilcoxon signed-rank test; data not shown) during MUS sessions (dMUS and iMUS) compared with the PBS session (p-values <0.05, Bonferroni-corrected post hoc test). This increase in session duration was attributable to arena wall bumping events, which usually entailed a recovery period before rats left the peripheral boundaries and moved again toward the goal. These observations indicate that

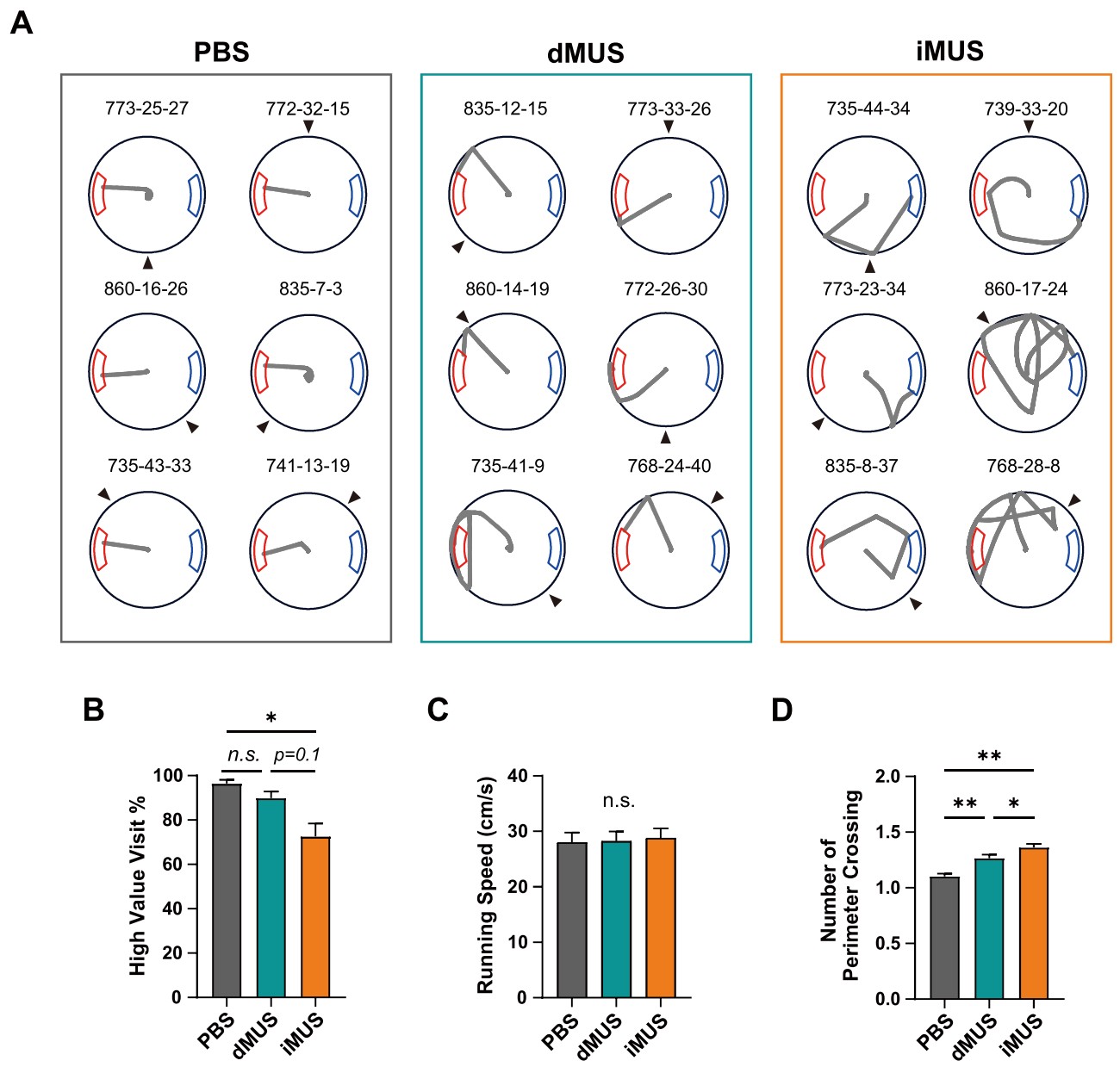

**Figure 5.** Changes in navigational pattern with each drug condition. (**A**) Sample trajectories in each drug condition. Black arrowheads indicate the start direction and the gray line shows the trajectory for each trial. Numbers above each trajectory indicate the identification numbers for rat, session, and trial. (**B**) Mean high-value zone visit percentage for each drug condition ($F_{(2,14)}$ = 10.56, p<0.01, one-way repeated-measures ANOVA; p=0.2 for PBS vs. dMUS, p<0.05 for PBS vs. iMUS, p=0.1 for dMUS vs. iMUS, Bonferroni-corrected post hoc test). Gray, green, and orange each indicate PBS, dMUS, and iMUS sessions, respectively. (**C**) Average running speed ($F_{(2,14)}$ = 0.99, p=0.37, one-way repeated-measures ANOVA). (**D**) Number of perimeter crossings ($F_{(1.16, 8.13)}$=1.34, p=0.29, one-way repeated-measures ANOVA with Greenhouse–Geisser correction; p<0.01 for PBS vs. dMUS, p<0.01 for PBS vs. iMUS, p<0.05 for dMUS vs. iMUS, Bonferroni-corrected post hoc test). For the PBS session, dPBS and iPBS sessions were first tested for significant differences between sessions; if they were not different, they were averaged to one PBS session for analysis purposes. The significance level was set at $\alpha$ = 0.05, and all error bars indicate SEMs (n=8). *p<0.05, **p<0.01.

inactivation of the iHP significantly impairs the rat's ability to effectively navigate to the higher-value reward zone in a VR environment without affecting goal-directedness or locomotor activity.

To determine how effectively rats traveled to the goal in each condition, we also quantified the errors made in each condition by assessing the number of perimeter crossings (*Figure 5D*). To avoid duplicate assessments, we only counted an event as a perimeter crossing when the rat crossed the perimeter boundary from inside to outside. Rats tended to make more errors in dMUS sessions

compared with controls, and errors were even more prevalent in iMUS sessions ($F_{(2,14)}$ = 18.59, p<0.001, one-way repeated-measures ANOVA; p=0.39 for dPBS vs. iPBS, Wilcoxon signed-rank test; p<0.01 for PBS vs. dMUS, p<0.01 in PBS vs. iMUS; p<0.05 for dMUS vs. iMUS, Bonferroni-corrected post hoc test). During PBS sessions, navigation was mostly precise, resulting in just one perimeter crossing. In the dMUS sessions, precision declined, but the rats were relatively successful in finding the high-value zone, with most trials being associated with a slightly increased number of perimeter crossings. In contrast, rats in the iMUS sessions failed to find the high-value zone. They seemed undirected, exhibiting a significantly increased number of perimeter crossings compared with the other two sessions. Taken together, these results indicate that iHP inactivation more severely damages normal goal-directed navigational patterns than dHP inactivation in our place-preference task.

## The iHP causes more damage to value-dependent spatial navigation than the dHP, which is important for navigational precision

To further differentiate among conditions, we examined DD and PCD – indices of the effectiveness and precision of navigation, respectively (*Figure 3*). We first investigated the distribution of DDs in all trials for all rats and calculated the resultant mean vector (*Figure 6A*). Note that dPBS and iPBS sessions were separately illustrated here for better visualization of changes in behavioral pattern for each subregion. Whereas DDs for both PBS sessions (dPBS and iPBS) were distributed relatively narrowly toward the high-value zone, those for dMUS sessions were more widely distributed, and their peak pointed away from the reward zone. In the case of the iHP-inactivation session, some DDs were even pointed toward the opposite side of the target goal zone (i.e., the low-value zone). Thus, the mean vectors from PBS sessions were relatively longer than those from MUS sessions. The mean vectors for PBS sessions also stayed within the range of the high-value zone, whereas those for MUS sessions pointed either toward the edge of the reward zone (dMUS) or the outside of the reward zone (iMUS).

We next quantitatively confirmed these observations, comparing the mean direction for each drug condition to determine how inactivation affected the accuracy of the body alignment of rats at departure (*Figure 6A*). A Watson–Williams test indicated that the mean angles of DDs in all four drug conditions (dPBS, iPBS, dMUS, and iMUS) for all rats significantly differed from each other ($F_{(3,1253)}$ = 7.78, p<0.001). Post hoc pairwise comparisons showed that inactivation of either the dHP or iHP significantly altered DDs compared with the PBS condition (p<0.05 for dPBS vs. dMUS; p<0.001 for iPBS vs. iMUS; p=0.66 for dPBS vs. iPBS; Watson–Williams test). Moreover, the mean DDs of dMUS and iMUS sessions were displaced from the center of the high-value zone compared with those of PBS sessions (i.e., dPBS and iPBS), suggesting that the rats did not accurately align themselves to the target reward zone at the time of departure. The mean vector of the iMUS session also appeared smaller than that of the other conditions, indicating a less concentrated distribution of DDs with iHP inactivation. Unfortunately, it was not possible to perform a statistical comparison between dMUS and iMUS because the DDs for the iMUS session were too dispersed to yield a mean vector with a sufficient length to compare directions between the two conditions (averaged mean vector length of dMUS and iMUS sessions <0.45; *Berens, 2009*).

The mean vector lengths for DDs were also significantly different among drug conditions ($F_{(2,14)}$ = 12.64, p<0.001, one-way repeated-measures ANOVA; p=0.55 for dPBS vs. iPBS, Wilcoxon signed-rank test) (*Figure 6B*), with a post hoc analysis showing a significant difference between the iMUS session and both PBS (p<0.01) and dMUS (p=0.05) sessions; however, no significant difference was found between PBS and dMUS sessions (p=0.24, Bonferroni-corrected post hoc test). The profound performance deficits in the iHP-inactivated condition were also confirmed by examining the DD-deviation from the target direction, defined as the center of the high-value zone (*Figure 6C*). Specifically, we found that DD-deviations were significantly different among drug conditions ($F_{(2,14)}$ = 13.37, p<0.001, one-way repeated-measures ANOVA; p=0.38 for dPBS vs. iPBS, Wilcoxon signed-rank test), with a Bonferroni-corrected post-hoc test revealing a significant increase in DD-deviation in the iMUS session compared with both the PBS session (p<0.01) and the dMUS session (p<0.05). Again, no significant difference was found between PBS and dMUS sessions (p=0.19). These results demonstrate that disruption of the dHP does not significantly affect the ability of rats to orient themselves effectively at departure to target the high-value reward zone. In contrast, inactivation of the iHP across all trials caused rats to depart the starting

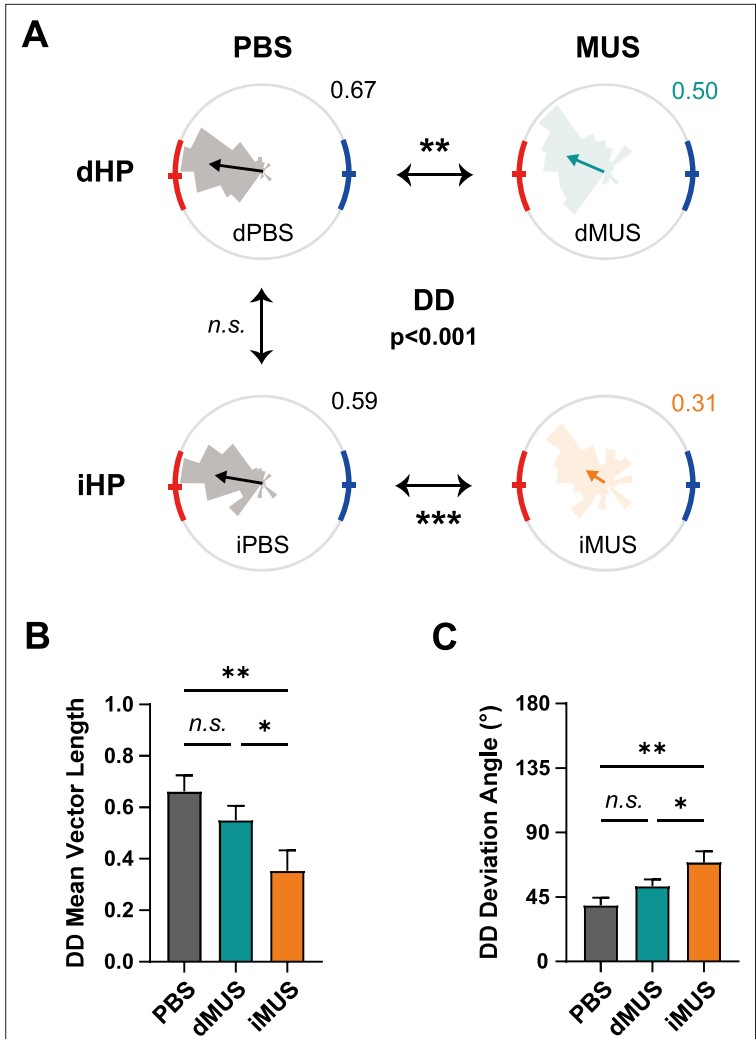

**Figure 6.** Dorsal hippocampal (dHP) and intermediate hippocampal (iHP) inactivation differentially affect efficient goal-directed navigation. (**A**) Grouped comparison of departing direction (DD) in each drug condition. Distributions of DDs in each drug condition (rose plots) and a comparison of their mean directions. Gray plots, PBS sessions; green plots, dHP inactivation; orange plots, iHP inactivation. Red and blue arcs indicate high- and low-value zones, respectively. Statistically significant differences in mean vectors, illustrated as arrows, are indicated with asterisks. The mean directions of all four conditions were first compared together ($F_{(3,1253)}$ = 7.78, p<0.001, Watson–Williams test); a post hoc pairwise comparison was subsequently applied if the average mean vector length of the two sessions was greater than 0.45 (p<0.05 for dPBS vs. dMUS; p<0.001 for iPBS vs. iMUS; p=0.66 for dPBS vs. iPBS; Watson–Williams test). The number on the upper-right side of the plot shows the length of the mean vector. (**B, C**) Changes in mean vector length ($F_{(2,14)}$ = 12.64, p<0.001, one-way repeated-measures ANOVA; p=0.24 for PBS vs. dMUS, p<0.01 for PBS vs. iMUS, p<0.05 for dMUS vs. iMUS, Bonferroni-corrected post hoc test) (**B**) and deviation angles from the high-value zone center ($F_{(2,14)}$ = 13.37, p<0.001, one-way repeated-measures ANOVA; p=0.19 for PBS vs. dMUS, p<0.01 for PBS vs. iMUS, p<0.05 for dMUS vs. iMUS) (**C**) of the DD in each drug session. Error bars indicate SEMs (n=8), and the significance level was set at α = 0.05. *p<0.05, **p<0.01, ***p<0.001.

location without strategically aligning to the scene and consequently failing to hit the target zone effectively and directly.

Next, we ran similar analyses for the PCD (*Figure 7*), also investigating the PCD distribution and its mean vector for each drug condition (*Figure 7A*). PCD distributions appeared similar to those for DD; the PCD distributions of PBS sessions were narrowly contained within the high-value reward zone, whereas those of MUS sessions were more dispersed and misaligned with the reward zone. Again, the PCD distribution of the iMUS session showed some occurrences near the low-value zone.

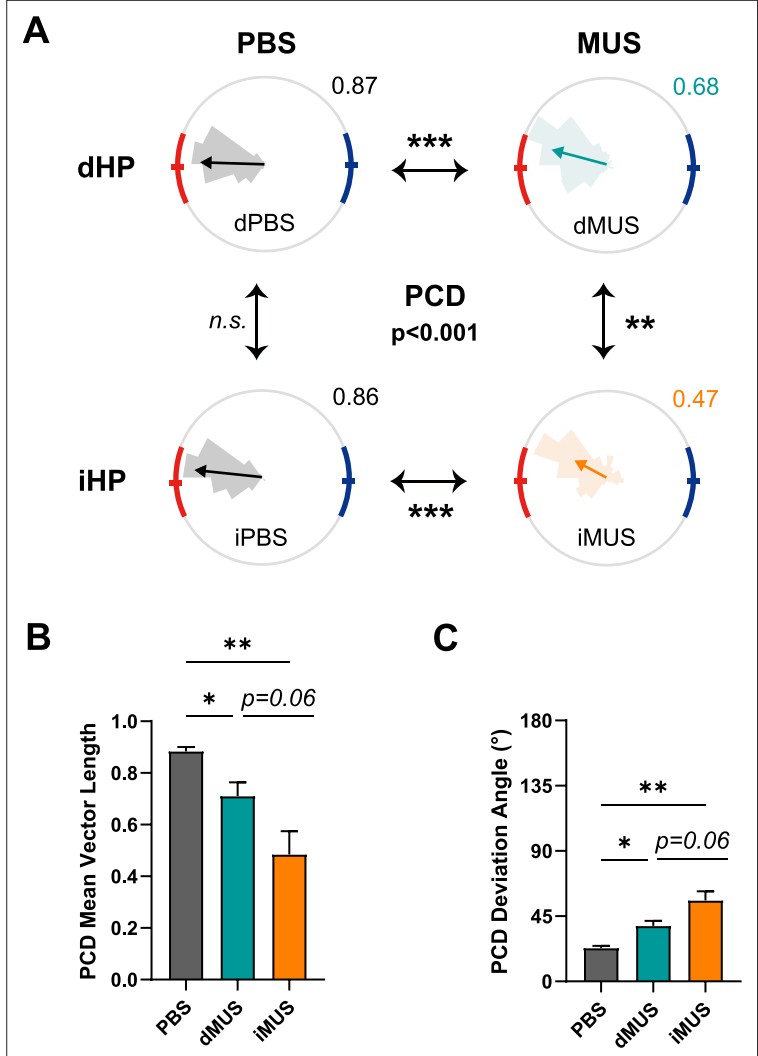

**Figure 7.** Precision of goal-directed navigation is more severely impaired with intermediate hippocampal (iHP) inactivation. (**A–C**) Same as *Figure 6*, except showing perimeter-crossing direction (PCD). (**A**) Grouped comparison of PCD in each drug condition.($F_{(3,1253)}$ = 16.22, p<0.001; p<0.001 for dPBS vs. dMUS, p<0.001 for iPBS vs. iMUS, p<0.01 for dMUS vs. iMUS, Watson-Williams test) . (**B**) Changes in mean vector length of the PCD in each drug condition ($F_{(2,14)}$ = 15.67, p<0.001, one-way repeated-measures ANOVA; p<0.05 for PBS vs. dMUS; p<0.01 for PBS vs. iMUS; p=0.06 for dMUS vs. iMUS, Bonferroni-corrected post hoc test). (**C**) Deviation angles from the high-value zone center of the PCD in each drug condition ($F_{(2,14)}$ = 17.24, p<0.001, one-way repeated-measures ANOVA; p<0.05 for PBS vs. dMUS, p<0.01 for PBS vs. iMUS; p=0.06 for dMUS vs. iMUS, Bonferroni-corrected post hoc test). Data are plotted as means ± SEMs (n=8), and the significance level was set at α = 0.05. *p<0.05, **p<0.01, ***p<0.001.

An examination of the resulting mean vectors using a Watson–Williams test revealed a significant difference in mean PCD angle in all sessions except for the comparison between the two PBS sessions ($F_{(3,1253)}$ = 16.22, p<0.001; p=0.08 for dPBS vs. iPBS). The mean PCD angle of the dMUS session was shifted toward the upper end of the high-value zone (p<0.001 for dPBS vs. dMUS), whereas that of the iMUS session was outside of the reward zone (p<0.001 for iPBS vs. iMUS). Notably, iHP inactivation resulted in more severe errors in finding the high-value zone than dHP inactivation (p<0.01 for dMUS vs. iMUS). Interestingly, with iHP inactivation, several PCDs were found near the low-value zone, an outcome that rarely occurred in other conditions. Considering the decreased percentage of high-value zone visits (*Figure 5*), some of these trials ended with the rat visiting the low-value zone, suggesting an impaired ability of the animal to perform goal-directed navigation strategically.

The PCD mean vector length was largest in the PBS condition, shortest in the iMUS condition, and intermediate in the dMUS condition ($F_{(2,14)}$ = 15.67, p<0.001, one-way repeated-measures ANOVA; p=0.55 for dPBS vs. iPBS, Wilcoxon signed-rank test) (*Figure 7B*). Unlike the mean vector length for DD, the PCD mean vector length differed between PBS and dMUS sessions, suggesting that wayfinding behavior was explicitly disrupted by dHP inactivation, albeit to a lesser extent compared with iHP inactivation (p<0.05 for PBS vs. dMUS; p<0.01 for PBS vs. iMUS; p=0.06 for dMUS vs. iMUS, Bonferroni-corrected post hoc test).

On the other hand, the PCD deviation angle from the center of the high-value zone increased in inverse order: smallest in the PBS condition and largest in the iMUS condition ($F_{(2,14)}$ = 17.24, p<0.001, one-way repeated-measures ANOVA; p=0.55 for dPBS vs. iPBS, Wilcoxon signed-rank test) (*Figure 7C*). Similar to the PCD mean vector length data, the significant increase in deviation angle after dHP inactivation indicates that dHP-inactivated rats failed to achieve fine spatial tuning toward the high-value zone compared with controls (p<0.05 for PBS vs. dMUS, Bonferroni-corrected post hoc test). iHP inactivation also resulted in less accurate navigation, including perimeter crossings – effects that were more severe than those caused by dHP inactivation (p<0.01 for PBS vs. iMUS; p=0.06 for dMUS vs. iMUS, Bonferroni-corrected post hoc test).

Overall, results based on the PCD measure revealed that dHP-inactivated rats showed decreased precision in arriving at the goal, as reflected in the significant deviation of their PCD from the high-value zone. The PCD distribution was also not as narrow as under control conditions. Notably, deficits in navigation performance were even more severe in rats with iHP inactivation, and their performance impairment was qualitatively different from that observed with dHP inactivation in terms of both efficiency and precision of navigation. Again, these results suggest that, while the dHP is essential for accurate wayfinding, the iHP is crucial for value-dependent navigation to the higher-reward location.

## Hippocampal inactivation does not impair cue-guided navigation or goal-directedness

After the drug injection stage, we trained five of the same rats used in the main task in a visual cue-guided navigation task to verify whether MUS inactivation of the hippocampus resulted in deficits in goal-directed navigation in general (*Figure 8Ai*). We used the same circular arena from the main

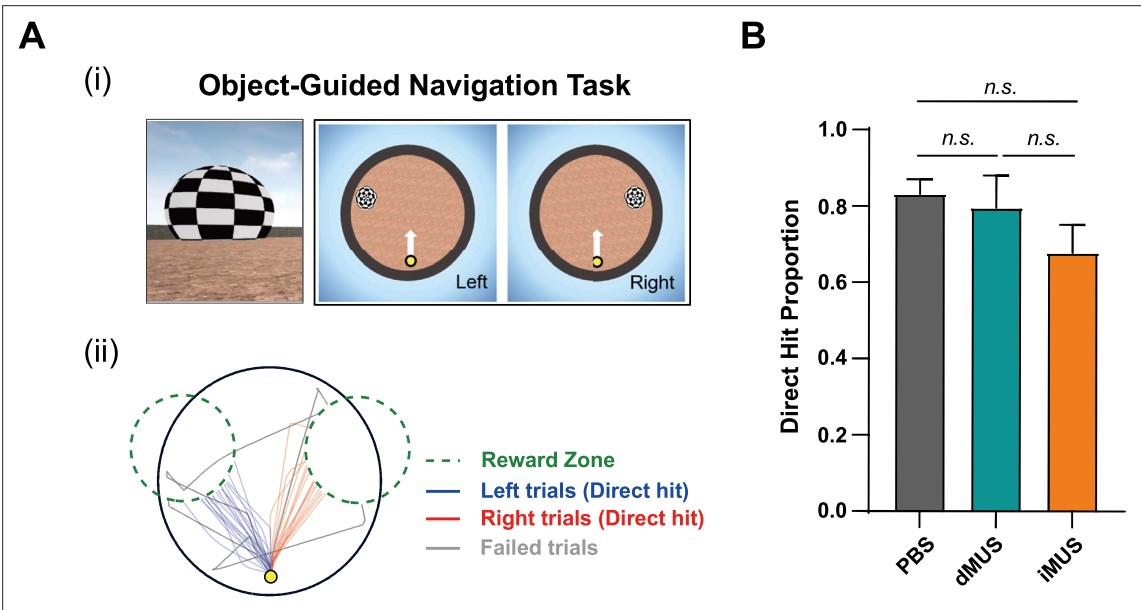

**Figure 8.** Goal-directedness and navigational capacity are unaffected by drug infusion. (**A**) Object-guided navigation task as a probe test. (**i**) A flickering object appeared on either the left ('Left trial') or right ('Right trial') side of the screen. The start location is marked with a yellow dot, with a white arrow indicating the start direction, which remained the same for both trial types. (**ii**) Example of trajectories in one session. Blue and red lines represent trajectories from left and right trials that directly arrived at the reward zones, whereas gray lines indicate failed trials. Green dashed lines denote reward zones. (**B**) Comparison of the proportion of each drug condition's direct hit trials (both left and right; $F_{(2,8)}$ = 1.60, p=0.26, one-way repeated-measures ANOVA). Error bars indicate SEMs (n=5), and the significance level was set at $\alpha$ = 0.05.

task but removed all allocentric visual landmarks. The rat was started from a fixed location near the periphery of the arena. As the trial started, a spherical visual landmark with a checkered pattern flickered on either the left or right side (pseudorandomized across trials) of the rat's starting position, serving as a beacon. When the rat arrived at the landmark area (see 'Materials and methods'), the connection between treadmill movement and the virtual environment stopped, and honey water was provided as a reward. The rewards provided by left and right reward zones were the same in terms of both quality and quantity.

In this version of the navigation task, the rat's navigation was simply guided by the visual beacon, a type of task that the literature suggests is not hippocampal-dependent (*Morris et al., 1986*; *Packard et al., 1989*). Rats learned the task rapidly. Specifically, it took 3 days on average for rats to reach the criterion of completing 40 trials with an excess travel distance of less than 0.1 m (see 'Materials and methods'). Moreover, examining their trajectories suggested that rats had no problem moving toward the visual landmark, whether it appeared on the left or right of the starting location (*Figure 8Aii*). Rats arrived at the reward zone directly in most trials ('Direct hit') but bumped into the arena wall in some trials. Given the presence of a strong visual landmark, which served as a beacon, trials in which the rat bumped the arena walls were considered failed trials (*Figure 8Aii*).

Finally, we applied the same drug injection schedule for the main task after the rats reached the abovementioned criterion. A one-way repeated-measures ANOVA revealed that the proportion of direct hit trials did not significantly differ across drug conditions, indicating no significant change in navigation precision when the goal was marked by the visual beacon ($F_{(2,8)}$ = 1.60, p=0.26; p=0.50 for dPBS vs. iPBS, Wilcoxon signed-rank test) (*Figure 8B*). These results also imply that no generic sensorimotor or motivational deficits were involved. Collectively, these observations confirm that MUS injections in the hippocampus do not alter the ability of the rat to move around freely in the VR environment in a goal-directed fashion when the hippocampus is not necessary for the task.

## Discussion

In the current study, we inactivated the dorsal or intermediate hippocampal region in rats performing a place-preference task in VR space to investigate the functional differentiation along the dorsoventral hippocampal axis during goal-directed navigation. Inactivation of the intermediate region, but not the dorsal region, of the hippocampus produced a marked reduction in the rat's ability to conduct strategic goal-directed navigation in the virtual space without affecting goal-directedness or locomotor ability. We further examined navigational quality by measuring the precision of scene alignment upon departure and by assessing the efficiency (i.e., directness) of travel to the target goal zone (i.e., higher-value zone) without bumping into walls on the arena boundaries. We found that dHP-inactivated rats were modestly, but significantly, impaired not only in precisely targeting the goal at the time of departure but also in effectively traveling to the goal zone, compared with controls. Importantly, however, the ability of these dHP-inactivated rats to head toward the higher-value zone in the VR environment was unimpaired. In contrast, iHP-inactivated rats were severely impaired in the initial targeting of the goal zone at the time of departure and traveled somewhat aimlessly in the VR environment compared with both controls and dHP-inactivated rats. Our findings suggest that the dHP is essential for finding the most effective travel path for precise spatial navigation and that the iHP is necessary for navigating the space in a value-dependent manner to achieve goals.

### Rats use allocentric visual scenes and landmarks to target the goal zone and adjust their paths accordingly during navigation in the VR environment

In the current paradigm, rats rotated the spherical treadmill counterclockwise immediately after the trial started at the VR arena's center, presumably to find the visual scene to guide them directly toward the goal zone (i.e., high-value zone) upon departure. This initial orientation of departure – or DD – seems critical in our task, as evidenced by the fact that, during training, rats that miscalculated the DD usually bumped into the wall and had to reorient themselves at various positions within the environment. Once the rats learned the task, they oriented themselves before leaving the start point by rotating the visual environment until they found the goal-associated visual scenes and then ran

straight toward the goal zone. These behavioral characteristics suggest that our task is heavily dependent on the rat's ability to use the allocentric reference frame of the visual environment.

Prior studies suggest that, in an environment where the directional information comes largely from allocentric visual cues, the spiking activity of place cells is significantly modulated by directional visual cues, a finding that holds in both real and virtual environments (*Acharya et al., 2016*; *Ravassard et al., 2013*; *Aronov and Tank, 2014*). In one of these studies (Acharya et al.), directional modulation of place cells was observed even during random foraging in the absence of a goal-directed memory task. Notably, this was also true for spatial view cells in nonhuman primates (*Rolls and O'Mara, 1995*; *Georges-François et al., 1999*). Although we did not record place cells in our study, hippocampal place cells could be predicted to exhibit directional firing patterns associated with the visual scenes along the periphery of the current VR environment. Because rats in the current study rotated the environment until they found the target-matching scene without leaving the center starting point, our VR task may be an ideal behavioral paradigm for examining the directional firing of place cells in future studies.

## Inactivating the dHP impairs navigational precision but does not affect place preference based on differential reward values

Our working model posits that the dHP represents a fine-scaled spatial map of an environment, in this case, a VR environment, that allows an animal to map its location precisely and choose the most efficient travel routes. Our experimental results support this model, demonstrating that dHP-inactivated rats deviated slightly, but significantly, from the ideal target heading at the time of departure (measured by DD), resulting in crossing the area boundary near the target goal zone. Nonetheless, it is important to note that dHP-inactivated rats in our study oriented themselves normally in the direction of the high-value reward zone at the time of departure, suggesting that the value-coding cognitive map and its use were intact and able to spatially guide the rats to the high-reward area in the absence of a functioning dHP. We argue that such intact place-preference performance with reasonable spatial navigation ability is supported by the iHP (presumably in connection with the vHP) in dHP-inactivated rats.

Whether the dHP represents value signals remains a matter of controversy. According to previous studies, place fields of the dHP seem to translocate to or accumulate near the location with motivational significance (e.g., reward zone), and where the strategic importance is higher (e.g., choice point in the T-maze) (*Hollup et al., 2001*; *Lee et al., 2006*; *Kennedy and Shapiro, 2009*; *Dupret et al., 2010*; *Ainge et al., 2011*; *Valenti et al., 2018*). For instance, the overrepresentation of the escape platform in a water maze – a location of high motivational significance – was observed in the neural firing patterns of place cells in the hippocampus (*Hollup et al., 2001*). In addition, Lee et al. reported that dHP place fields gradually translocate toward the goal arm of a continuous T-maze (*Lee et al., 2006*), and Dupret and colleagues suggested a goal-directed reorganization of hippocampal place fields based on an experimental paradigm in which reward locations were changed daily (*Dupret et al., 2010*). Such accumulation of spatial firing is not restricted to the goal location, as place fields recorded from the dHP were reported to be unevenly distributed near the start box and the choice point of a T-maze (*Kim et al., 2012*). One potential explanation for the discrepancy between our study and studies that reported apparent valence-dependent signals in the dHP could be that the dHP processes motivational and strategic significance (from the perspective of task demand), which is not always the same as the reward. Significance might include task demand, such as a change between random and directed search of reward (*Markus et al., 1995*), or a change in a significant environment stimulus from which the goal location needs to be calculated (*Gothard et al., 1996*). However, none of these were changed in our experimental paradigm, which might explain why dHP inactivation did not affect place-preference behavior.

Another possibility is that the dHP responds only to a more radical change in value, such as the presence or absence of reward, but not to different amounts of the same reward. Indeed, hippocampal neuronal activity does not show an explicit response to reward value in rats trained to visit arms of a plus-maze in descending order of reward amount (*Tabuchi et al., 2003*). Moreover, when the reward is unexpectedly altered to a less preferred one, thus decreasing motivational significance, place cells in the dHP remain mostly unchanged (*Jin and Lee, 2021*). These results suggest that the dHP is not crucial to maintaining value preference, a finding in line with the observed absence of an effect on place preference after dHP inactivation in our study. A recent study in which mice were

trained to associate a particular odor with an appetitive outcome and distinguish it from the non-rewarded odor suggested that the dHP is responsible for stimulus identity, not saliency (*Biane et al., 2023*). This might be another possible interpretation of our dHP results since we used the same honey water reward for both reward zones.

## The iHP may contain a value-associated cognitive map with reasonable spatial resolution for value-based navigation

iHP-inactivated rats showed poor goal-directed navigation, characterized by misalignment of their departing orientation with the goal zone and arrival points that were often far removed from the goal zone compared with the same rats under both control and dHP-inactivated conditions. Particularly, rats changed their heading directions during the navigation when they were not confident with the location of the higher reward, resulting in a less efficient route to the goal location. Rats showing this type of behavior tended to hit the perimeter of the arena first before correcting their routes. Therefore, when considered together with DD, our PCD measure could tell that the rats not hitting the goal zone directly after departure were impaired in orienting themselves to the target zone accurately from the start, not in maintaining the correct heading direction to the goal zone at the start location.

Although there is still a possibility that the levels of expression of GABA-A receptors might be different along the longitudinal axis of the hippocampus, these results support our working model that the iHP is critical for representing a value-associated cognitive map of the environment. iHP-inactivated rats, presumably unable to utilize such a value-representing map, could not strategically plan and organize their behaviors to target the high-value area in the current study. Consequently, the fine-grained spatial map present in the dHP may be of little use without the guidance of the value-associated map in the iHP, accounting for the poor navigational performance of iHP-inactivated rats. The value-associated cognitive map in the iHP may still show reasonable spatial specificity, as evidenced by the larger, but still specifically located, place fields in the iHP compared with the dHP (*Jin and Lee, 2021*).

The involvement of the iHP in spatial value association has been reported or implicated in several studies. For example, Bast and colleagues reported that rapid place learning is disrupted by removing the iHP and vHP, even when the dHP remains undamaged (*Bast et al., 2009*). On the other hand, if the iHP is spared but the dHP and vHP are removed by lesioning, rats in a water maze test quickly learn a new platform location normally. Moreover, a change in reward value induced an immediate global remapping response and a greater overrepresentation of the reward zone with a higher value in iHP neurons (*Jin and Lee, 2021*). Another recent study by Jarzebowski et al. focused on how hippocampal place cells change their firing patterns during the learning process for several sets of changing reward locations (*Jarzebowski et al., 2022*). The results from this study suggest that, in the iHP, the same place cells persistently fire across different reward locations, thus tracking the changes in reward locations.

Anatomically, the iHP is in an ideal position to represent associations between a space and its value by intrahippocampal connections from both the dHP and vHP (*Tao et al., 2021*; *Swanson et al., 1978*). Importantly, the vHP is known to receive much heavier projections from value-processing subcortical areas, such as the amygdala and VTA, compared with the upper two-thirds of the hippocampus (*Krettek and Price, 1977*; *Swanson et al., 1978*; *Pikkarainen et al., 1999*; *Felix-Ortiz and Tye, 2014*; *Gasbarri et al., 1994*). Thus, although the iHP also receives afferent projections from these areas, it is highly likely that the vHP plays a crucial role in the value-related representation of the iHP. Notably, both the amygdala and VTA are known to be involved in processing palatability information (*Tye and Janak, 2007*; *Fontanini et al., 2009*; *Chen et al., 2020*), and the amygdala has a subpopulation of neurons dedicated to encoding positive values (*Kim et al., 2016*; *Beyeler et al., 2016*). These anatomical studies support our working model of the iHP in integrating place-value information.

It is worth noting that the iHP sends direct projections to the mPFC, which is thought to be involved in behavioral control and action (*Hoover and Vertes, 2007*; *Liu and Carter, 2018*). Our experimental paradigm required rats to choose and navigate toward one of two reward zones with different values, a task structure that must demand active cognitive control, presumably by the mPFC in collaboration with the hippocampus. It is also possible that inactivation of the iHP prevents the transfer of the dHP's

spatial information to the mPFC via the iHP, which may explain why iHP inactivation produces severe deficits in goal-directed navigation in the current task. Based on these findings, we propose a working model in which the iHP associates spatial value information with the cognitive map of the dHP and sends value-associated spatial information to the mPFC, which translates the space-value-integrated representation into action (*Bast, 2011*; *Bast et al., 2009*).

## Limitations

We tested the differential functions of the hippocampal subregions in the long axis, dHP and iHP, by inactivating each subregion during goal-directed navigation. The subregional inactivation allowed us to compare the differences in navigational patterns directly between the drug conditions within subjects. However, our study includes only behavioral results and further mechanistic explanations as to the processes underlying the behavioral deficits require physiological investigations at the cellular level. Neurophysiological recordings during VR task performance could answer, for example, the questions such as whether the value-associated map in the iHP is built upon the map inherited from the dHP or it is independently developed in the iHP. Also, although our observations and behavioral data strongly suggest that rats rely on allocentric visual scenes in the VR environment instead of a single or limited set of landmarks, it is still difficult to prove experimentally whether rats used the cognitive map of the virtual arena to find the high-value zone or they had an alternative strategy to find the goal.

# Materials and methods

## Subjects

Eight male Long–Evans rats (8 weeks old) were housed individually under a 12 hr light/dark cycle in a temperature- and humidity-controlled environment. Rats were food-restricted to maintain ~80% of their free-feeding weight, but water was provided ad libitum. The experimental protocol (SNU-200504-3-1) complied with the guidelines of the Institutional Animal Care and Use Committee of Seoul National University. Based on our prior studies (*Park et al., 2017*; *Yoo and Lee, 2017*; *Lee et al., 2014*), the sample size of our study was set to the least number to achieve the necessary statistical power in the current within-subject study design for ethical commitments and practical considerations (i.e., relatively long training periods).

## 2D VR system

We established our own VR environment consisting of a circular arena surrounded by multiple landmarks using a game engine (Unreal Engine [UE] 4.14.3; Epic Games, Inc, USA; *Figure 1A and B*). The VR environment was presented via five adjacent LCD monitors covering 270° of the visual field. Rats were body-restrained at the top of a spherical treadmill, and a silicone-coated Styrofoam ball with 400 mm diameter was placed on multiple ball bearings. Rats could move their heads freely; body jackets were used to anchor their positions, limiting their body movements to a 120° range. As rats rolled the treadmill, their movement was recorded by three rotary encoders (DBS60E-BGFJD1024; Sick, Inc, Germany) attached to the treadmill surface. The signal from the encoders was then sent to the computer and synchronized with the movement in the virtual environment via an Arduino interface board (Arduino Leonardo; Arduino, Italy) and MATLAB R2021a (MathWorks, USA). A licking port was placed in front of the rats and moved in association with their body movement. It was maintained in a retracted position but was extended toward the snout by a linear motor (L16-R; Actuonix Motion Devices, Canada); an infrared sensor (FD-S32; Panasonic Industry, Japan) detected rats' tongues to record licking behavior. When rats arrived at either reward zone, the solenoid valve (VA212-3N; Aonetech, Republic of Korea), controlled by the UE via the Arduino interface (Arduino UNO; Arduino, Italy), dispensed honey water as a reward. The amount of honey water dispensed for high-value and low-value zones was maintained at a ratio (in drops) of 12:2, with 12 µl per drop.

## Behavioral paradigm

After several days of handling, rats were moved to the VR apparatus and trained to roll the treadmill to navigate the virtual environment ('Shaping'; *Figure 4B*). In this session, rats had to reach a flickering checkerboard-shaped sphere randomly spawned on a circular arena (1.6 m in diameter) to obtain a honeywater reward. After rats had completed more than 60 trials on two consecutive days, they were assumed to have adapted to navigating freely. They were moved to pre-surgical training ('Pre-training') – a 2D VR version of the place-preference task. For the pre-training session, rats were required to find hidden reward zones using the surrounding scene, including various landmarks, such as houses, mountains, and arches, on the same circular arena from the shaping session. The start position was located at a fixed point in the center of the arena, and reward zones were located at the east and west sides of the circular platform; reward zones were positioned at a slight distance from the arena wall to prevent rats from employing a thigmotaxis strategy. Therefore, the shortest path length between the start position and the reward zone was 0.62 m. A trial started with a heading in one of six start directions, pseudorandomly chosen, and ended when the rat arrived at either reward zone. Pre-surgical training criteria were defined by the number of trials (60 trials in 40 min), high-value zone visit percentage (>75%), and average excess travel distance (<0.6 m). If a rat successfully achieved training criteria 2 days in a row, it received cannula implantation surgery.

After the surgery, rats were allowed 1 week of recovery ('Recovery') and then were moved to postsurgical training ('Post-training'). During post-training, rats were tested on the same place-preference task until they achieved the same criteria as pre-surgical training, except that the trial number was reduced to 40 and the average excess travel distance was reduced to less than 1 m. This point marked the beginning of the drug injection stage; four rats received their initial injection in the dHP, and the other four rats were injected first in the iHP to counterbalance the injection order ('Place-Preference Task').

## Object-guided navigation task

After completing drug injections, we trained five of the eight rats from the main task for an object-guided navigation task to investigate whether drug infusion caused any motor- or motivation-related impairments ('Probe'; *Figure 4B*). Note the smaller sample size in the object-guided navigation task. This was because the task was later added to the study design. In this task, the rat simply had to find and navigate toward a flickering object; because there was no need for the rat to use the surrounding scene to locate the reward, this probe test was hippocampus-independent. For the probe test, the rat started from the south of the arena; concurrently, a flickering checkerboard-shaped sphere appeared on either the left or right side of the screen. When the rat arrived at the reward zone (i.e., a 0.4-m-radius circle surrounding the object), the visual stimulus stopped, a honeywater reward was given, and the trial ended. No landmarks surrounded the circular arena to distinguish the environment from that in the main task. The criterion for training included the completion of 40 trials with less than 0.1 m of mean excess travel distance, calculated as the shortest path length between the start location and the reward zone; a time limit of 60 s was also imposed. The drug infusion schedule from the place-preference task was then repeated, at which point rats were sacrificed for histological procedures.

## Surgery

After rats reached pre-surgical training criteria, they were implanted with four commercial cannulae (P1 Technologies, USA), bilaterally targeting the dHP and the iHP, enabling within-subject comparisons in performance between dHP inactivation (dMUS) and iHP inactivation (iMUS) conditions (*Figure 4A*). Animals were first anesthetized with an intraperitoneal injection of sodium pentobarbital (Nembutal, 65 mg/kg), then their heads were fixed in a stereotaxic frame (Kopf Instruments, USA). Isoflurane (0.5–2% mixed with 100% oxygen) was used to maintain anesthesia throughout the surgery. The cannula tips targeted approximately the upper blades of the dentate gyrus of both regions (AP –3.8 mm, ML ±2.6 mm, DV –2.7 mm for the dHP; AP –6.0 mm, ML ±5.6 mm, DV –3.2 mm with a 10° tilt for the iHP) to inactivate each subregion effectively. The cannula, consisting of a 26-gauge guide cannula coupled with a 33-gauge dummy cannula, was fixed to the target location by several skull screws and bone cement. After the surgery, ibuprofen syrup was orally administered for pain relief, and the animal was kept in an intensive care unit overnight.

## Drug infusion

For drug injection, the rat was first anesthetized with isoflurane. Then, 0.3–0.5 µl of either PBS or muscimol (MUS; 1 mg/ml, dissolved in saline) was infused into each hemisphere via a 33-gauge injection cannula at an injection speed of 0.167 µl/min, based on our previous study (*Lee et al., 2014*; *Kim et al., 2012*). The injection cannula and dummy cannula extended 1 mm below the tip of the guide cannula. The injection cannula was left in place for 1 min after completing the drug infusion to ensure stable diffusion of the drug. Then, it was slowly removed from the guide cannula and replaced by the dummy cannula. The rat was kept in a clean cage to recover from anesthesia completely and monitored for side effects for 20 min, then was moved to the VR apparatus for behavioral testing. If the rat showed any side effect, particularly sluggishness or aggression, we reduced the drug injection amount in the rat by 0.1 µl until we found the dosage with which there was no visible side effect. As a result, five of the rats received 0.4 µl, two received 0.3 µl, and one received 0.5 µl.

## Histology

After completing the probe test, animals were sacrificed by inhalation of an overdose of $CO_2$. Rats were then transcardially perfused, first with PBS, administered with a syringe, and then with a 4% v/v formaldehyde solution, delivered using a commercial pump (Masterflex Easy-Load II Pump; Cole-Parmer, USA). The brain was extracted and placed in a 4% v/v formaldehyde–30% sucrose solution at 4°C until it sank to the bottom of the container. After gelatin embedding, the brain was sectioned at 40 µm using a microtome (HM430; Thermo Fisher Scientific, USA), and sections were mounted on subbed slide glasses for Nissl staining.

## Statistical analysis

Data were statistically analyzed using custom programs written in MATLAB R2021a (MathWorks), Prism 9 (GraphPad, USA), and SPSS (IBM, USA). Statistical significance was determined using the Wilcoxon signed-rank test and one-way repeated-measures analysis of variance (RM ANOVA) followed by a Bonferroni post hoc test. Although most of our statistics were based on the nonparametric tests for the relatively small sample size (n = 8), we used the parametric RM ANOVA for comparing three groups (i.e., PBS, dMUS, and iMUS) because it is the most commonly known and widely used statistical test in such comparison. However, we also performed statistical test with the alternatives for reference, and the statistical significances were not changed with any of the results. For directional analysis, Kuiper's test and Watson–Williams test were used. However, the latter test was considered inapplicable for the mean angle when the average mean vector length between two samples was less than 0.45 (*Berens, 2009*). The significance level was set at α = 0.05, and all error bars indicate the standard error of means (SEMs).

## Acknowledgements

This work was supported by the National Research Foundation of Korea (2019R1A2C2088799, 2021R1A4A2001803, 2022M3E5E8017723) and the Global Ph.D. Fellowship program (2019H1A2A1073456). We thank Heesoo Oh for his assistance in behavioral training.

## Additional information

### Funding

| Funder | Grant reference number | Author |
| --- | --- | --- |
| National Research Foundation of Korea | 2019R1A2C2088799 | Inah Lee |
| National Research Foundation of Korea | 2021R1A4A2001803 | Inah Lee |
| National Research Foundation of Korea | 2022M3E5E8017723 | Inah Lee |

| Funder | Grant reference number | Author |
|---|---|---|
| National Research Foundation of Korea | 2019H1A2A1073456 | Hyeri Hwang |

The funders had no role in study design, data collection and interpretation, or the decision to submit the work for publication.

## Author contributions

Hyeri Hwang, Conceptualization, Data curation, Software, Formal analysis, Visualization, Writing – original draft, Writing – review and editing; Seung-Woo Jin, Resources, Software, Methodology; Inah Lee, Conceptualization, Resources, Formal analysis, Supervision, Funding acquisition, Validation, Investigation, Visualization, Methodology, Writing – original draft, Project administration, Writing – review and editing

## Author ORCIDs

Hyeri Hwang http://orcid.org/0009-0000-7929-6403
Seung-Woo Jin http://orcid.org/0000-0001-5364-5433
Inah Lee https://orcid.org/0000-0003-3760-4257

## Ethics

This study was performed in strict accordance with the recommendations in the Guide for the Care and Use of Laboratory Animals of the Seoul National University. All of the animals were handled according to approved institutional animal care and use committee (IACUC) protocol (SNU-200504-3-1) of the Seoul National University.

Reviewer #1 (Public review): https://doi.org/10.7554/eLife.97114.3.sa1
Reviewer #2 (Public review): https://doi.org/10.7554/eLife.97114.3.sa2
Reviewer #3 (Public review): https://doi.org/10.7554/eLife.97114.3.sa3
Author response https://doi.org/10.7554/eLife.97114.3.sa4

# Additional files

## Supplementary files

• MDAR checklist

## Data availability

The behavioral data and codes used in this study can be accessed freely through https://doi.org/10.5281/zenodo.12593588.

The following dataset was generated:

| Author(s) | Year | Dataset title | Dataset URL | Database and Identifier |
|---|---|---|---|---|
| Hwang H, Jin SW, Lee I | 2024 | hhwang28/Hwang-et-al.-eLife-2024: Hwang et al., eLife 2024_v2 | https://doi.org/10.5281/zenodo.12593588 | Zenodo, 10.5281/zenodo.12593588 |

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
