## [Editor Report · eLife assessment]

The authors report **solid** evidence for a **valuable** set of findings in rats performing a new virtual place-preference task. Temporary pharmacological inhibition targeting the dorsal or intermediate hippocampus disrupted navigation to a goal location in the task, and functional inhibition of the intermediate hippocampus was more detrimental than functional inhibition of the dorsal hippocampus. The work provides novel insights into functional differentiation along the dorsal-ventral axis of the hippocampus.

---

## [Referee Report · Reviewer #1 (Public review)]

Summary:

The manuscript examines the contribution of dorsal and intermediate hippocampus to goal-directed navigation in a wide virtual environment where visual cues are provided by the scenery on the periphery of a wide arena. Among a choice of 2 reward zones located near the arena periphery, rats learn to navigate from the center of the arena to the reward zone associated with the highest reward. Navigation performance is largely assessed from the rats' body orientation when they leave the arena center and when they reach the periphery, as well as the angular mismatch between reward zone and the site rats reach the periphery. Muscimol inactivation of dorsal and intermediate hippocampus alters rat navigation to the reward zone, but the effect was more pronounced for the inactivation of intermediate hippocampus, with some rat trajectories ending in the zone associated with the lowest reward. Based on these results, the authors suggest that the intermediate hippocampus is critical especially for navigating to the highest reward zone.

Strengths:

- The authors developed an effective approach to study goal-directed navigation in a virtual environment where visual cues are provided by the peripheral scenery.

- In general, text is clearly written and the figures are well designed and relatively straightforward to interpret, even without reading the legends.

- An intriguing result, which would deserve to be better investigated and/or discussed, was that rats tended to rotate always in the counterclockwise direction. Could this be because of a hardware bias making it easier to turn left, some aspect of the peripheral landscape, or a natural preference of rats to turn left that is observable (or reported) in real environment?

- Another interesting observation, which would also deserved to be addressed in the discussion, is the fact that dHP/iHP inactivations produced to some extent consistent shifts in departing and peripheral crossing directions. This is visible from the distributions in Figures 6 and 7, which still show a peak under muscimol inactivation, but this peak is shifted to earlier angles than the correct ones. Such change is not straightforward to interpret, unlike the shortening of the mean vector length.

Maybe rats under muscimol could navigate simply using association of reward zone with some visual cues in the peripheral scene, in brain areas other than the hippocampus, and therefore stopped their rotation as soon as they saw the cues, a bit before the correct angle. While with their hippocampus intact, rats could estimate precisely the spatial relationship between the reward zone and visual cues.

Weaknesses:

- I am not sure that the differential role of dHP and iHP for navigation to high/low reward locations is supported by the data. The current results could be compatible with iHP inactivation producing a stronger impairment on spatial orientation than dHP inactivation, generating more erratic trajectories that crossed by chance the second reward zone.

To make the point that iHP inactivation affects disambiguation of high and low reward locations, the authors should show that the fraction of trajectories aiming at the low reward zone is higher than expected by chance. Somehow we would expect to see a significant peak pointing toward the low reward zone in the distribution of Figures 6-7.

Review of revised manuscript

The experimental paradigm and analyses are interesting/novel and generate some intriguing phenomena such as the animals' preference for counterclockwise rotation and the stereotypical trajectory shifts induced by muscimol inactivation. Understanding better the underlying mechanisms of these phenomena and the navigational strategies involved in this apparatus will be important in the future for correctly interpreting inactivation experiments.

The idea of a differential effect of dMUS and iMUS was toned down in the abstract and other parts of the manuscript, such that the claims now better match the data.

---

## [Referee Report · Reviewer #2 (Public review)]

Summary:

The aim of this paper was to elucidate the role of the dorsal HP and intermediate HP (dHP and iHP) in value-based spatial navigation through behavioral and pharmacological experiments using a newly developed VR apparatus. The authors inactivated dHP and iHP by muscimol injection and analyzed the differences in behavior. The results showed that dHP was important for spatial navigation, while iHP was critical for both value judgments and spatial navigation. The present study developed a new sophisticated behavioral experimental apparatus and proposed a behavioral paradigm that is useful for studying value-dependent spatial navigation. In addition, the present study provides important results that support previous findings of differential function along the dorsoventral axis of the hippocampus.

---

## [Referee Report · Reviewer #3 (Public review)]

Summary:

The authors established a new virtual reality place preference task. On the task, rats, which were body-restrained on top of a moveable Styrofoam ball and could move through a circular virtual environment by moving the Styrofoam ball, learnt to navigate reliably to a high-reward location over a low-reward location, using allocentric visual cues arranged around the virtual environment.

The authors also showed that functional inhibition by bilateral microinfusion of the GABA-A receptor agonist muscimol, which targeted the dorsal or intermediate hippocampus, disrupted task performance. The impact of functional inhibition targeting the intermediate hippocampus was more pronounced than that of functional inhibition targeting the dorsal hippocampus.

Moreover, the authors demonstrated that the same manipulations did not significantly disrupt rats' performance on a virtual reality task that required them to navigate to a spherical landmark to obtain reward, although there were numerical impairments in the main performance measure and the absence of statistically significant impairments may partly reflect a small sample size (see under Weaknesses, point 3.).

Overall, the study established a new virtual-reality place preference task for rats and established that performance on this task requires the dorsal to intermediate hippocampus. It also established that task performance is more sensitive to the same muscimol infusion when the infusion was applied to the intermediate hippocampus, compared to the dorsal hippocampus. The authors suggest that these differential effects of muscimol infusions reflect that dorsal hippocampus is responsible for 'precise' spatial navigation and intermediate hippocampus for place-value associations, but this idea remains to be tested by further studies. In their first revision to the paper, the authors toned down this claim, but I still think it would be good to consider more explicitly potential alternative explanations for the differential effects of dorsal and intermediate muscimol infusions (see under Weaknesses, point 2.).

Strengths:

(1) The authors established a new place preference task for body-restrained rats in a virtual environment and, using temporary pharmacological inhibition by intra-cerebral microinfusion of the GABA-A receptor agonist muscimol, showed that task performance requires dorsal to intermediate hippocampus.

(2) These findings extend our knowledge about place learning tasks that require dorsal to intermediate hippocampus and add to previous evidence that the intermediate hippocampus may be more important than other parts of the hippocampus, including the dorsal hippocampus, for goal-directed navigation based on allocentric place memory.

(3) The hippocampus-dependent task may be useful for future recording studies examining how hippocampal neurons may support behavioral performance based on place information.

Weaknesses:

(1) The new findings do not strongly support the authors' suggestion that dorsal hippocampus is responsible for precise spatial navigation and intermediate hippocampus for place-value associations (e.g., final sentence of the first paragraph of the Discussion). The authors base this claim on differential effects of the dorsal and intermediate hippocampal muscimol infusions on different performance measures on the virtual reality place preference task. More specifically, dorsal hippocampal muscimol infusion significantly increased perimeter crossings and perimeter crossing deviations, whereas other measures of task performance are not significantly changed, including departure direction and visits to the high-value location. However, these statistical outcomes offer only limited evidence that dorsal hippocampal infusion specifically affected the perimeter crossing, without affecting the other measures. Numerically the pattern of infusion effects is quite similar across these various measures: intermediate hippocampal infusions markedly impaired these performance measures compared to vehicle infusions, and the values of these measures after dorsal hippocampal muscimol infusion were between the values in the intermediate hippocampal muscimol and the vehicle condition (Figs 5-7). In my opinion, these findings could reflect that dorsal and intermediate hippocampus play distinct roles, as suggested by the authors, but the findings are also consistent with the suggestion that intermediate hippocampal muscimol had a quantitatively stronger, but qualitatively similar effect to dorsal hippocampal muscimol. However, I am largely content with the authors acknowledging within the paper that their suggestion would need to be confirmed by additional studies.

Moreover, I do not find it clearly described in the paper which distinct aspects of navigation the departure direction and perimeter crossing deviation measures capture. The authors suggest that departure direction and perimeter crossing deviation are indices of the navigational efficiency and precision of navigation, respectively (e.g., from p. 7, line 195). However, both departure direction and perimeter crossing deviation measure how accurate/precise, in other words 'close to the target', the rat's navigation is. Efficiency of navigation may rather be reflected by the path length taken (a measure that was not reported). It appears to me that a key difference between the two measures is that departure direction measures the rat's direction towards the goal at the beginning of the rat's navigational path, whereas perimeter crossing deviation measures this further toward the end of the navigational path. This would suggest that departure direction may depend more on directional orienting mechanisms early on in the rat's journey, whereas perimeter crossing deviation may also depend on fine-grained place recognition as the rat approaches the goal. Given the fine-grained place representations in the dorsal hippocampus, the latter may, therefore, depend more on the dorsal hippocampus than the former. I think this would fit with the authors' suggestion 'that the dHP represents a fine-scaled spatial map of an environment' (p. 18, first line). If the authors agree with my interpretation of the different measures, they may consider clarifying this in the Results and Discussion sections.

(2) The claim that the different effects of intermediate and dorsal hippocampal muscimol infusions reflect different functions of intermediate and dorsal hippocampus rests on the assumption that both manipulations inhibit similar volumes of hippocampal tissue to a similar extent, but at different levels along the dorso-ventral axis of the hippocampus. However, this is not a foregone conclusion (e.g., drug spread may differ depending on the infusion site or drug effects may differ due to differential distribution or efficiency of GABA-A receptors), and the authors do not provide direct evidence for this assumption. Therefore, an alternative account of the weaker effects of dorsal compared to intermediate hippocampal muscimol infusions on place-preference performance is that the dorsal infusions affect less hippocampal volume or less markedly inhibit neurons within the affected volume than the intermediate infusions (e.g., due to different drug spread following dorsal and intermediate infusions or due to different distribution or effectiveness of GABA-A receptors in dorsal and intermediate hippocampus). I would recommend that the authors explicitly state this limitation in the limitations section of the Discussion. In their response to my original comments, the authors argue that it is unlikely that muscimol exerts stronger effects in intermediate compared to dorsal hippocampus, based on the finding that in vitro paired pulse inhibition is reduced in ventral compared to dorsal hippocampal slices (Papatheodoropoulos et al., 2002). However, this claim is not strongly supported by the in vitro paired-pulse inhibition findings. First, these findings relate to differences between dorsal and ventral hippocampus, whereas differences between dorsal and intermediate hippocampus were not investigated. Second, reduced paired pulse inhibition may not necessarily reflect reduced GABA-A receptor expression/efficiency (which would be likely to reduce muscimol effects), but may also reflect reduced GABAergic input, which would not be expected to reduce muscimol effects.

(3) It is good that the authors included a comparison/control experiment using a spherical beacon-guided navigation task, to examine the specific psychological mechanisms disrupted by the hippocampal manipulations. However, the sample size for the comparison experiment (n = 5 rats) was lower than for the main study n = 8 rats, and the data shown in Fig. 8 suggest that the comparison task may be affected by the hippocampal manipulations similarly to the place-preference task, albeit less markedly. This effect may well have been significant if the same sample size had been used as in the main experiment. Therefore, I would recommend that the authors acknowledge this limitation in the Discussion (perhaps, in the Limitation section).

---

## [Author Response]

The following is the authors’ response to the original reviews.

**Public Reviews:**

**Reviewer #1 (Public Review):**
Summary:The manuscript examines the contribution of the dorsal and intermediate hippocampus to goal-directed navigation in a wide virtual environment where visual cues are provided by the scenery on the periphery of a wide arena. Among a choice of 2 reward zones located near the arena periphery, rats learn to navigate from the center of the arena to the reward zone associated with the highest reward. Navigation performance is largely assessed from the rats' body orientation when they leave the arena center and when they reach the periphery, as well as the angular mismatch between the reward zone and the site rats reach the periphery. Muscimol inactivation of the dorsal and intermediate hippocampus alters rat navigation to the reward zone, but the effect was more pronounced for the inactivation of the intermediate hippocampus, with some rat trajectories ending in the zone associated with the lowest reward. Based on these results, the authors suggest that the intermediate hippocampus is critical, especially for navigating to the highest reward zone.Strengths:-The authors developed an effective approach to study goal-directed navigation in a virtual environment where visual cues are provided by the peripheral scenery.- In general, the text is clearly written and the figures are well-designed and relatively straightforward to interpret, even without reading the legends.- An intriguing result, which would deserve to be better investigated and/or discussed, was that rats tended to rotate always in the counterclockwise direction. Could this be because of a hardware bias making it easier to turn left, some aspect of the peripheral landscape, or a natural preference of rats to turn left that is observable (or reported) in a real environment?

Thank you for the insightful question. As the reviewer mentioned, the counterclockwise rotation behavior was intriguing and unexpected. To answer the reviewer’s question properly, we examined whether such stereotypical turning behavior appeared before the rats acquired the task rule and reward zones in the pre-surgical training phase of the task. Data from the last day of shaping and the first day of the pre-surgical main task day showed no significant difference in the number of trials in which the first body-turn was either clockwise or counterclockwise, suggesting that the rats did not have a bias toward a specific side (p=0.46 for Shaping; p=0.76 for the Main task, Wilcoxon signed-rank test). These results excluded the possibility that there was something in the apparatus's hardware that made the rats turn only to the left. Also, since we used the same peripheral landscape for the shaping and main task, we could assume that the peripheral landscape did not cause movement bias.

**Author response image 1. sa4fig1:** 

Although it remains inconclusive, we have noticed that some prior studies alluded to a phenomenon similar to this issue, framed as the topic of lateralization or spatial preference by comparing left and right biases. For example, Wishaw et al. (1992) suggested that there was natural lateralization in rats (“Most of the rats displayed either a strong right limb bias or a strong left limb bias.”) but no dominance to a specific side. Andrade et al. (2001) also claimed that “83% of Wistar rats spontaneously showed a clear preference for left or right arms in the T-maze.” However, to the best of our knowledge, there has been no direct evidence that rats have a dominant natural preference only to one side.

Therefore, while the left-turning behavior remains an intriguing topic for further investigation, we find it difficult to pinpoint the reason behind the behavior in the current study. However, we would like to emphasize that this behavior did not interrupt testing our hypothesis. Nonetheless, we agree with the reviewer’s point that the counterclockwise rotation needs to be discussed more, so we revised the manuscript as follows:

“To rule out the potential effect of hardware bias or any particular aspect of peripheral landscape to make rats turn only to one side, we measured the direction of the first body-turn in each trial on the last day of shaping and the first day of the main task (i.e., before rats learned the reward zones). There was no significant difference between the clockwise and counterclockwise turns (p=0.46 for shaping, p=0.76 for main task; Wilcoxon signed-rank test), indicating that the stereotypical pattern of counterclockwise body-turn appeared only after the rats learned the reward locations.” (p.6)

- Another interesting observation, which would also deserve to be addressed in the discussion, is the fact that dHP/iHP inactivations produced to some extent consistent shifts in departing and peripheral crossing directions. This is visible from the distributions in Figures 6 and 7, which still show a peak under muscimol inactivation, but this peak is shifted to earlier angles than the correct ones. Such change is not straightforward to interpret, unlike the shortening of the mean vector length.Maybe rats under muscimol could navigate simply by using the association of reward zone with some visual cues in the peripheral scene, in brain areas other than the hippocampus, and therefore stopped their rotation as soon as they saw the cues, a bit before the correct angle. While with their hippocampus is intact, rats could estimate precisely the spatial relationship between the reward zone and visual cues.

We agree with the possibility suggested by the reviewer. However, although not described in the original manuscript, we performed several different control experiments in a few rats using various visual stimulus manipulations to test how their behaviors change as a result. One of the experiments was the landmark omission test, where one of the landmarks was omitted. The landmark to be made disappear was pseudorandomly manipulated on a trial-by-trial basis. We observed that the omission of one landmark, regardless of its identity, did not cause a specific behavioral change in finding the reward zones, suggesting that the rats were not relying on a single visual landmark when finding the reward zone.

**Author response image 2. sa4fig2:** 

Therefore, it is unlikely that rats used the spatial relationship between the reward zone and a specific visual cue to solve the task in our study. However, the result was based on an insufficient sample size (n = 3), not permitting any meaningful statistical testing. Thus, we have now updated this information in the manuscript as an anecdotal result as follows:

“Additionally, to investigate whether the rats used a certain landmark as a beacon to find the reward zones, we conducted the landmark omission test as a part of control experiments. Here, one of the landmarks was omitted, and the landmark to be made disappear was pseudorandomly manipulated on a trial-by-trial basis. The omission of one landmark, regardless of its identity, did not cause a specific behavioral change in finding the reward zones, suggesting that the rats were not relying on a single visual landmark when finding the reward zones. The result can be reported anecdotally only because of an insufficient sample size (n = 3), not permitting any meaningful statistical testing.” (p.9)

Weaknesses:-I am not sure that the differential role of dHP and iHP for navigation to high/low reward locations is supported by the data. The current results could be compatible with iHP inactivation producing a stronger impairment on spatial orientation than dHP inactivation, generating more erratic trajectories that crossed by chance the second reward zone.To make the point that iHP inactivation affects the disambiguation of high and low reward locations, the authors should show that the fraction of trajectories aiming at the low reward zone is higher than expected by chance. Somehow we would expect to see a significant peak pointing toward the low reward zone in the distribution of Figures 6-7.

We thank the reviewer for the valuable comments. We agree that it is difficult to rigorously distinguish the loss of value representation from spatial disorientation in our experiment. Since the trial ended once the rat touched either reward zone, it was difficult to specify whether they intended to arrive at the location or just moved randomly and arrived there by chance. Moreover, it is possible that the drug infusion did not completely inactivate the iHP but only partially did so.

To investigate this issue further, we checked whether the distribution of the departure direction (DD) differed between the trials in which rats initially headed north (NW, N, NE) and south (SE, S, SW) at the start. In the manuscript, we demonstrated that DD aligned with the high-value zone, indicating that the rat remembered the scenes associated with the high-value zone (p.8). Based on the rats’ characteristic counterclockwise rotation, the reward zone rats would face first upon starting while heading north would be the high-value zone. On the other hand, the rat would face the low-value reward zone when starting while heading south. In this case, normal rats would inhibit leaving the start zone and rotate further until they face the high-value zone before finally departing the start location. If the iHP inactivation caused a more severe impairment in spatial orientation but not in value representation, it is likely that the iHP-inactivated rats in both north- and south-starting trials would behave similarly with the dHP-inactivated rats, but producing a larger deviation from the high-value zone. However, if the iHP inactivation affected the disambiguation of high and low reward locations, north and south-starting trials would show different DD distributions.

The circular plots shown below are the DD distributions of dMUS and iMUS. We could see that when they started facing north, iHP-inactivated rats still aligned themselves towards the high-value zone and thus remained spatially oriented, similar to the dHP inactivation session. However, in the south-starting trials, the DD distribution was completely different from the north-starting trials; the rats failed in body alignment towards the high-value zone. Instead, they departed the start point while heading south in most trials. This pattern was not seen in dMUS sessions, even in their south-starting trials, illustrating the distinct deficit caused by iHP inactivation. Additionally, most of the rats with iHP inactivation visited the low-value zone more in south-headed starting trials than in the north-headed trials, except for one rat.

**Author response image 3. sa4fig3:** 

Furthermore, we would like to clarify that we do not limit the effect of iHP inactivation to the impairment in distinguishing the high and low reward zones. It is possible that iHP inactivation resulted in the loss of a global value-representing map, leading to the impairment in distinguishing both reward zones from other non-rewarded areas in the environment. Figures 6 and 7 implicated this possibility by showing that the peaks are not restricted only to the reward zones. Unfortunately, we cannot rigorously address this in the current study because of the limitations of our experimental design mentioned above.

Nonetheless, we agree with the reviewer that this limitation needs to be addressed, so we now added how the current study needs further investigation to clarify what causes the behavioral change after the iHP inactivation in the Limitations section (p.21).

**Reviewer #2 (Public Review):**
Summary:The aim of this paper was to elucidate the role of the dorsal HP and intermediate HP (dHP and iHP) in value-based spatial navigation through behavioral and pharmacological experiments using a newly developed VR apparatus. The authors inactivated dHP and iHP by muscimol injection and analyzed the differences in behavior. The results showed that dHP was important for spatial navigation, while iHP was critical for both value judgments and spatial navigation. The present study developed a new sophisticated behavioral experimental apparatus and proposed a behavioral paradigm that is useful for studying value-dependent spatial navigation. In addition, the present study provides important results that support previous findings of differential function along the dorsoventral axis of the hippocampus.Strengths:The authors developed a VR-based value-based spatial navigation task that allowed separate evaluation of "high-value target selection" and "spatial navigation to the target." They were also able to quantify behavioral parameters, allowing detailed analysis of the rats' behavioral patterns before and after learning or pharmacological inactivation.Weaknesses:Although differences in function along the dorsoventral axis of the hippocampus is an important topic that has received considerable attention, differences in value coding have been shown in previous studies, including the work of the authors; the present paper is an important study that supports previous studies, but the novelty of the findings is not that high, as the results are from pharmacological and behavioral experiments only.

We appreciate the reviewer's insightful comments. In response, we would like to emphasize that a very limited number of studies investigated the function of the intermediate hippocampus, especially in spatial memory tasks. We tested the differential functions of the dorsal and intermediate hippocampus using a within-animal design and used reversible inactivation manipulation (i.e., muscimol injection) to prevent potential compensation by other brain regions when using irreversible manipulation techniques (i.e., lesion). Also, very few studies have analyzed the navigation trajectories of animals as closely as in the current study. We emphasize the novelty of our study by comparing it with prior studies, as shown below in Table 1.

**Author response table 1. sa4table1:** Comparison of our study with those from prior studies.

	Targetregions	Withinanimalcomparison	Manipulation	Valuedifference	Detailed,analysis,on,trajectory
Jarzebowski et al.,2022	Dorsal,intermediate	No	No	No	No
Bast et al., 2009	Dorsal,intermediate,ventral	No	Lesion	No	No
Moser et al., 1995	Dorsal,ventral	No	Lesion	No	No
De Hoz et al., 2003	Dorsal,ventral	No	Lesion	No	No
Ferbinteanu andMcDonald, 2001	Dorsal,ventral	No	Lesion	No	Yes(headingangle)
De Saint Blanquat etal., 2013	Dorsal,intermediate-ventral	No	Reversibleinactivation(Muscimol)	No	No
Tabuchi et al., 2003	Dorsal,ventral	Yes	No	Yes	No
Jin and Lee, 2021(previous study)	Dorsal,intermediate	Yes	No	Yes	No
Hwang et al., 2024(current study)	Dorsal,intermediate	Yes	Reversibleinactivation(Muscimol)	Yes	Yes

Moreover, to the best of our knowledge, the current manuscript is the first to investigate the hippocampal subregions along the long axis in a VR environment using a hippocampal-dependent spatial memory task. Nonetheless, we agree that the current study has a limitation as a behavior-only experiment. We now have added a comment on how other techniques, such as electrophysiology, would develop our findings in the Limitation section (p.21).

**Reviewer #3 (Public Review):**
Summary:The authors established a new virtual reality place preference task. On the task, rats, which were body-restrained on top of a moveable Styrofoam ball and could move through a circular virtual environment by moving the Styrofoam ball, learned to navigate reliably to a high-reward location over a low-reward location, using allocentric visual cues arranged around the virtual environment.The authors also showed that functional inhibition by bilateral microinfusion of the GABA-A receptor agonist muscimol, which targeted the dorsal or intermediate hippocampus, disrupted task performance. The impact of functional inhibition targeting the intermediate hippocampus was more pronounced than that of functional inhibition targeting the dorsal hippocampus.Moreover, the authors demonstrated that the same manipulations did not significantly disrupt rats' performance on a virtual reality task that required them to navigate to a spherical landmark to obtain reward, although there were numerical impairments in the main performance measure and the absence of statistically significant impairments may partly reflect a small sample size (see comments below).Overall, the study established a new virtual-reality place preference task for rats and established that performance on this task requires the dorsal to intermediate hippocampus. They also established that task performance is more sensitive to the same muscimol infusion (presumably - doses and volumes used were not clearly defined in the manuscript, see comments below) when the infusion was applied to the intermediate hippocampus, compared to the dorsal hippocampus, although this does not offer strong support for the authors claim that dorsal hippocampus is responsible for accurate spatial navigation and intermediate hippocampus for place-value associations (see comments below).Strengths:(1) The authors established a new place preference task for body-restrained rats in a virtual environment and, using temporary pharmacological inhibition by intra-cerebral microinfusion of the GABA-A receptor agonist muscimol, showed that task performance requires dorsal to intermediate hippocampus.(2) These findings extend our knowledge about place learning tasks that require dorsal to intermediate hippocampus and add to previous evidence that, for some place memory tasks, the intermediate hippocampus may be more important than other parts of the hippocampus, including the dorsal hippocampus, for goal-directed navigation based on allocentric place memory.(3) The hippocampus-dependent task may be useful for future recording studies examining how hippocampal neurons support behavioral performance based on place information.Weaknesses:(1) The new findings do not strongly support the authors' suggestion that the dorsal hippocampus is responsible for accurate spatial navigation and the intermediate hippocampus for place-value associations.The authors base this claim on the differential effects of the dorsal and intermediate hippocampal muscimol infusions on different performance measures. More specifically, dorsal hippocampal muscimol infusion significantly increased perimeter crossings and perimeter crossing deviations, whereas dorsal infusion did not significantly change other measures of task performance, including departure direction and visits to the high-value location. However, these statistical outcomes offer only limited evidence that dorsal hippocampal infusion specifically affected the perimeter crossing, without affecting the other measures. Numerically the pattern of infusion effects is quite similar across these various measures: intermediate hippocampal infusions markedly impaired these performance measures compared to vehicle infusions, and the values of these measures after dorsal hippocampal muscimol infusion were between the values in the intermediate hippocampal muscimol and the vehicle condition (Figures 5-7). Moreover, I am not so sure that the perimeter crossing measures really reflect distinct aspects of navigational performance compared to departure direction and hit rate, and, even if they did, which aspects this would be. For example, in line 316, the authors suggest that 'departure direction and PCD [perimeter crossing deviation] [are] indices of the effectiveness and accuracy of navigation, respectively'. However, what do the authors mean by 'effectiveness' and 'accuracy'? Accuracy typically refers to whether or not the navigation is 'correct', i.e. how much it deviates from the goal location, which would be indexed by all performance measures.So, overall, I would recommend toning down the claim that the findings suggest that the dorsal hippocampus is responsible for accurate spatial navigation and the intermediate hippocampus for place-value associations.

The reviewer mentioned that the statistical outcomes offer limited evidence as the dHP inactivation results were always positioned between the results of the iHP inactivation and controls. However, we would like to emphasize that, projecting to each other, the two subregions are not completely segregated anatomically. It is highly likely this is also true functionally and there should be some overlap in their roles. Considering such relationships between the dHP and iHP, it could be natural to see an intermediate effect after inactivating the dHP, and that is why we focused on the “magnitude” of behavioral changes after inactivation instead of complete dissociation between the two subregions in our manuscript. Unfortunately, because of the nature of the drug infusion study, further dissociation would be difficult, requiring further investigation with different experimental techniques, such as physiological examinations of the neural firing patterns between the two regions. We mentioned this caveat of the current study in the Limitations as follows:

“However, our study includes only behavioral results and further mechanistic explanations as to the processes underlying the behavioral deficits require physiological investigations at the cellular level. Neurophysiological recordings during VR task performance could answer, for example, the questions such as whether the value-associated map in the iHP is built upon the map inherited from the dHP or it is independently developed in the iHP.” (p.21)

Regarding the reviewer’s comment on the meaning of measuring the perimeter crossing directions, we would like to draw the reviewer’s attention to the individual trajectories during the iMUS sessions described in Figure 5. Particularly when they were not confident with the location of the higher reward, rats changed their heading directions during the navigation, which resulted in a less efficient route to the goal location. Rats showing this type of behavior tended to hit the perimeter of the arena first before correcting their routes toward the goal zone. In contrast, rats showing effective navigation hardly bumped into the wall or perimeter before hitting the goal zone. Thus, their PCDs matched DDs almost always. When considered together with DD, our PCD measure could tell whether rats not hitting the goal zone directly after departure were impaired in either maintaining the correct heading direction to the goal zone at the start location or orienting themselves to the target zone accurately from the start. Our results suggest that the latter is the case. We included the relevant explanation in the Discussion section as follows:

“Particularly, rats changed their heading directions during the navigation when they were not confident with the location of the higher reward, resulting in a less efficient route to the goal location. Rats showing this type of behavior tended to hit the perimeter of the arena first before correcting their routes. Therefore, when considered together with DD, our PCD measure could tell that the rats not hitting the goal zone directly after departure were impaired in orienting themselves to the target zone accurately from the start, not in maintaining the correct heading direction to the goal zone at the start location.” (p.19)

Nonetheless, we agree with the reviewer that the term ‘accuracy’ might be confusing with performance accuracy, so we replaced the term with ‘precision’ throughout the manuscript, referring to the precise targeting of the reward zones.

(2) The claim that the different effects of intermediate and dorsal hippocampal muscimol infusions reflect different functions of intermediate and dorsal hippocampus rests on the assumption that both manipulations inhibit similar volumes of hippocampal tissue to a similar extent, but at different levels along the dorso-ventral axis of the hippocampus. However, this is not a foregone conclusion (e.g., drug spread may differ depending on the infusion site or drug effects may differ due to differential expression of GABA-A receptors in the dorsal and intermediate hippocampus), and the authors do not provide direct evidence for this assumption. Therefore, a possible alternative account of the weaker effects of dorsal compared to intermediate hippocampal muscimol infusions on place-preference performance is that the dorsal infusions affect less hippocampal volume or less markedly inhibit neurons within the affected volume than the intermediate infusions. I would recommend that the authors briefly consider this issue in the discussion. Moreover, from the Methods, it is not clear which infusion volume and muscimol concentration were used for the different infusions (see below, 4.a.), and this must be clarified.

We appreciate these insightful comments from the reviewer and agree that we do not provide direct evidence for the point raised by the reviewer. To the best of our knowledge, most of the behavioral studies on the long axis of the hippocampus did not particularly address the differential expression of GABA-A receptors along the axis. We could not find any literature that specifically introduced and compared the levels of expression of GABA-A receptors or the diffusion range of muscimol in the intermediate hippocampus to the other subregions. However, we found that Sotiriou et al. (2005) made such comparisons with respect to the expression of different GABA-A receptors. They concluded that the dorsal and ventral hippocampi have different levels of the GABA-A receptor subtypes. The a1/b2/g2 subtype was dominant in the dorsal hippocampus, while the a2/b1/g2 subtype was prevalent in the ventral hippocampus. Sotiriou and colleagues also mentioned the lower affinity of GABA-A receptor binding in the ventral hippocampus, and this result is consistent with the Papatheodoropoulos et al. (2002) study that showed a weaker synaptic inhibition in the ventral hippocampus compared to the dorsal hippocampus. Papatheodoropoulos et al. speculated differences in GABA receptors as one of the potential causes underlying the differential synaptic inhibition between the dorsal and ventral hippocampal regions. Based on these findings, the same volume of muscimol is more likely to cause a more severe effect on the ventral hippocampus than the dorsal hippocampus. Therefore, we do not believe that the less significant changes after the dorsal hippocampal inactivation were induced by the expression level of GABA-A receptors. Additionally, we have demonstrated in our previous study that muscimol injections in the dorsal hippocampus impair performance to the chance level in scene-based behavioral tasks (Lee et al., 2014; Kim et al., 2012).

Nonetheless, we mentioned the possibility of differential muscimol expressions between the two target regions. Following the suggestion of the reviewer, we now included this information in the Discussion as follows:

“Although there is still a possibility that the levels of expression of GABA-A receptors might be different along the longitudinal axis of the hippocampus, …” (p.20)

Regarding the drug infusion volume and concentration, we included these details in the Methods. Please see our detailed response to 4.a. below.

(3) It is good that the authors included a comparison/control study using a spherical beacon-guided navigation task, to examine the specific psychological mechanisms disrupted by the hippocampal manipulations. However, as outlined below (4.b.), the sample size for the comparison study was lower than for the main study, and the data in Figure 8 suggest that the comparison task may be affected by the hippocampal manipulations similarly to the place-preference task, albeit less markedly. This would raise the question as to which mechanisms that are common to the two tasks may be affected by hippocampal functional inhibition, which should be considered in the discussion.

The sample size for the object-guided navigation task was smaller because we initially did not plan the experiment, but later in the study decided to conduct the control test. Therefore, the object-guided navigation task was added to the study design after finishing the first three rats, resulting in a smaller sample size than the place preference task. We included this detail in the manuscript, as follows:

“Note the smaller sample size in the object-guided navigation task. This was because the task was later added to the study design.” (p.24)

Regarding the mechanism behind the two different tasks, we did not perform the same heading direction analysis here as in the place preference task because the two tasks have different characteristics such as task complexity. The object-guided navigation task is somewhat similar to the visually guided (or cued) version of the water maze task, which is widely known as hippocampal-independent (Morris et al., 1986; Packard et al., 1989; also see our descriptions on p.15). Therefore, we would argue that the two tasks (i.e., place preference task and object-guided navigation task) used in the current manuscript do not share neural mechanisms in common. Additionally, we confirmed that several behavioral measurements related to motor capacity, such as travel distance and latency, along with the direct hit proportion provided in Figure 8, did not show any statistically significant changes across drug conditions.

4. Several important methodological details require clarification:a. Drug infusions (from line 673):- '0.3 to 0.5 μl of either phosphate-buffered saline (PBS) or muscimol (MUS) was infused into each hemisphere'; the authors need to clarify when which infusion volume was used and why different infusion volumes were used.

We thank the reviewer for carefully reading our manuscript. We were cautious about side effects, such as suppressed locomotion or overly aggressive behavior, since the iHP injection site was close to the ventricle. We were keenly aware that the intermediate to ventral hippocampal regions are sensitive to the drug dosage from our previous experiments. Thus, we observed the rat’s behavior for 20 minutes after drug injection in a clean cage. We started from 0.5 μl, based on our previous study, but if the injected rat showed any sign of side effects in the cage, we stopped the experiment for the day and tried with a lower dosage (i.e., 0.4 μl first, then 0.3 μl, etc.) until we found the right dosage under which the rat did not show any side effect. This procedure is necessary because cannula tip positions are slightly different from rat to rat. When undergoing this procedure, five out of eight rats received 0.4 μl, two received 0.3 μl, and one received 0.5 μl. Still, there was no significant difference in performance, including the high-value visit percentage, departing and perimeter crossing directions, across all dosages. This information is now added in the Methods section as follows:

“If the rat showed any side effect, particularly sluggishness or aggression, we reduced the drug injection amount in the rat by 0.1 ml until we found the dosage with which there was no visible side effect. As a result, five of the rats received 0.4 ml, two received 0.3 ml, and one received 0.5 ml.” (p.25)

- I could not find the concentration of the muscimol solution that was used. The authors must clarify this and also should include a justification of the doses used, e.g. based on previous studies.

Thank you for the suggestion. We used the drug concentration of 1mg/ml, which was adapted from our previous muscimol study (Lee et al., 2014; Kim et al., 2012). The manuscript is now updated, as follows:

“…or muscimol (MUS; 1mg/ml, dissolved in saline) was infused into each hemisphere via a 33-gauge injection cannula at an injection speed of 0.167 ml/min, based on our previous study (Lee et al., 2014; Kim et al., 2012).” (p.25)

- Please also clarify if the injectors and dummies were flush with the guides or by which distance they protruded from the guides.

The injection and dummy cannula both protruded from the guide cannula by 1 mm, and this information is now added to the Methods section, as follows:

“The injection cannula and dummy cannula extended 1 mm below the tip of the guide cannula.” (p.25)

b. Sample sizes: The authors should include sample size justifications, e.g. based on considerations of statistical power, previous studies, practical considerations, or a combination of these factors. Importantly, the smaller sample size in the control study using the spherical beacon-guided navigation task (n = 5 rats) limits comparability with the main study using the place-preference task (n = 8). Numerically, the findings on the control task (Figure 8) look quite similar to the findings on the place-preference task, with intermediate hippocampal muscimol infusions causing the most pronounced impairment and dorsal hippocampal muscimol infusions causing a weaker impairment. These effects may have reached statistical significance if the same sample size had been used in the place-preference study.

We set the current sample size for several reasons. First, based on our previous studies, we assumed that eight, or more than six, would be enough to achieve statistical power in a “within-animal design” study. Also, considering the ethical commitments, we tried to keep the number of animals used in the study to the least. Last, our paradigm required very long training periods (3 months on average per animal), so we could not increase the sample size for practical reasons. Regarding the reasons for the smaller sample size for the object-guided navigation task, please see the previous response to 3 above. The manuscript is now revised as follows:

“Based on our prior studies (Park et al., 2017; Yoo and Lee, 2017; Lee et al., 2014), the sample size of our study was set to the least number to achieve the necessary statistical power in the current within-subject study design for ethical commitments and practical considerations (i.e., relatively long training periods).” (p.22)

c. Statistical analyses: Why were the data of the intermediate and dorsal hippocampal PBS infusion conditions averaged for some of the analyses (Figure 5; Figure 6B and C; Figure 7B and C; Figure 8B) but not for others (Figure 6A and Figure 7A)?

The reviewer is correct that we only illustrated the separate dPBS and iPBS data for Figures 6A and 7A. Since the directional analysis is the main focus of the current manuscript, we tried to provide better visualization and more detailed examples of how the drug infusion changed the behavioral patterns between the PBS and MUS conditions in each region. Except for the visualization of DD and PCD, we averaged the PBS sessions to increase statistical power, as described in p.9. We added a detailed description of the reasons for illustrating dPBS and iPBS data separately in the manuscript, as follows:

“Note that dPBS and iPBS sessions were separately illustrated here for better visualization of changes in the behavioral pattern for each subregion.” (p.12)

**Reviewing Editor (Recommendations For The Authors):**
The strength of evidence rating in the assessment is currently noted as "incomplete." This can be improved following revisions if you amend your conclusions in the paper, including in the title and abstract, such that the paper's major conclusions more closely match what is shown in the Results.

Following the suggestions of the reviewing editor, we have mentioned the caveats of our study in the Limitations section of our revised manuscript (p.21). In addition, the manuscript has been revised so that the conclusions in the paper match more closely to the experimental results as can been seen in some of the relevant sentences in the abstract and main text as follows:

“Inactivation of both dHP and iHP with muscimol altered efficiency and precision of wayfinding behavior, but iHP inactivation induced more severe damage, including impaired place preference. Our findings suggest that the iHP is more critical for value-dependent navigation toward higher-value goal locations.” (Abstract; p.2)

“Whereas inactivation of the dHP mainly affected the precision of wayfinding, iHP inactivation impaired value-dependent navigation more severely by affecting place preference.” (p.5)

“The iHP causes more damage to value-dependent spatial navigation than the dHP, which is important for navigational precision” (p.12)

However, we haven’t changed the title of the manuscript as it carries what we’d like to deliver in this study accurately.

**Reviewer #1 (Recommendations For The Authors):**
- What were the dimensions of the environment? What distance did rats typically run to reach the reward zone? A scale bar would be helpful in Figure 1.

We used the same circular arena from the shaping session, which was 1.6 meters in diameter (p.23), and the shortest path between the start location and either reward zone was 0.62 meters. We revised the manuscript for clarification as follows:

“For the pre-training session, rats were required to find hidden reward zones…, on the same circular arena from the shaping session.” (p.23)

“Therefore, the shortest path length between the start position and the reward zone was 0.62 meters.” (p.23)

We also added a scale bar in Figure 1C for a better understanding.

- Line 169: "The scene rotation plot covers the period from the start of the trial to when the rat leaves the starting point at the center and the departure circle (Figure 2B)."The sentence is unclear. Maybe it should be "... from the start of the trial to when the rat leaves the departure circle”.

The sentence has been revised following the reviewer's suggestion. (p.7)

- Line 147: "First, they learned to rotate the spherical treadmill counterclockwise to move around in the virtual environment (presumably to perform energy-efficient navigation)."It is not clear from this sentence if rats naturally preferred the counterclockwise direction or if the counterclockwise direction was a task requirement.

We now clarified in our revised manuscript that it was not a task requirement to turn counterclockwise, as follows:

“First, although it was not required in the task, they learned to rotate the spherical treadmill counterclockwise…” (p.6)

- Line 149: "Second, once a trial started, but before leaving the starting point at the center, the animal rotated the treadmill to turn the virtual environment immediately to align its starting direction with the visual scene associated with the high-value reward zone."The sentence is unclear. Maybe "Second, once a trial started, the animal rotated the treadmill immediately to align its starting direction with the visual scene associated with the high-value reward zone.”

We have updated the description following the suggestion. (p.6)

**Reviewer #2 (Recommendations For The Authors):**
- There are some misleading descriptions of the conclusion of the results in this paper. In this study, the functions of (a) selection of high-value target and (b) spatial navigation to the target were assessed in the behavioral experiments. The results of the pharmacological experiments showed that dHP inactivation impaired (b) and iHP inactivation impaired both (a) and (b) (Figures 5 B & D). However, the last sentence of the abstract states that dHP is important for the functions of (a) and iHP for (b). There are several other similar statements in the main text. Since the separation of (a) and (b) is an important and original aspect of this study, the description should clearly show the conclusion that dHP is important for (a) and iHP is important for both (a) and (b).Related to the above, the paragraph title in the Discussion "The iHP may contain a value-associated cognitive map with reasonable spatial resolution for goal-directed navigation (536-537)" is also somewhat misleading: "with reasonable resolution for goal-directed behavior" seems to reflect the results of an object-guided navigation task (Figure 8). However, the term "goal-directed behavior" is also used for value-dependent spatial navigation (i.e., the main task), which causes confusion. I would like to suggest clarifying the wording on this point.

First, we need to correct the reviewer’s statement regarding our descriptions of the results. As the reviewer mentioned, our results indicated that the dHP inactivation impaired (b) but not (a), while the iHP inactivation impaired both (a) and (b). Regarding the iHP inactivation result, we focused on the impairment of (a) since our aim was to investigate spatial-value association in the hippocampus. Also, it was more likely that (a) affected (b), but not the other way, because (a) remained intact when (b) was impaired after dHP inactivation. We emphasized this difference between dHP and iHP inactivation, which was (a). Therefore, we mentioned in the last sentence of the abstract that the dHP is important for (b), which is the precision of spatial navigation to the target location, and the iHP is critical for (a).

Moreover, we would like to clarify that we were not referring to the object-guided navigation task in Figure 8 in the phrase ‘with a reasonable spatial resolution for goal-directed navigation.’ Please note that the object-guided navigation task did not require fine spatial resolution to find the reward. The phrase instead referred to the dHP inactivation result (Figure 5 and 6), where the rats could find the high-value zone even with dHP inactivation, although the navigational precision decreased. Nonetheless, we agree with the reviewer for the confusion that the title might cause, so now have updated the title as follows:

“The iHP may contain a value-associated cognitive map with reasonable spatial resolution for value-based navigation” (p.19)

- As an earlier study focusing on the physiology of iHP, Maurer et al, Hippocampus 15:841 (2005) is also a pioneering and important study, and I suggest citing it.

Thank you for the suggestion. We included the Maurer et al. (2005) study in the Introduction section as follows:

“…Specifically, there is physiological evidence that the size of a place field becomes larger as recordings of place cells move from the dHP to the vHP (Jung et al., 1994; Maurer et al., 2005; Kjelstrup et al., 2008; Royer et al., 2010).” (p.4)

- One of the strengths of this paper is that we have developed a new control system for the VR navigation task device, but I cannot get a very detailed description of this system in the Methods section. Also, no information about the system control has been uploaded to GitHub. I would suggest adding a description of the manufacturer, model number, and size of components, such as a rotary encoder and ball, and information about the software of the control system, with enough detail to allow the reader to reconstruct the system.

We have now added detailed descriptions of the VR system in the Methods section (see “2D VR system). (p.22)

**Reviewer #3 (Recommendations For The Authors):**
(1) Some comments on specific passages of text:Lines 87 to 89: 'Surprisingly, beyond the recognition of anatomical divisions, little is known about the functional differentiation of subregions along the dorsoventral axis of the hippocampus. Moreover, the available literature on the subject is somewhat inconsistent.'I would recommend to rephrase these statements. Regarding the first statement, there is substantial evidence for functional differentiation along the dorso-ventral axis of the hippocampus (e.g., see reviews by Moser and Moser, 1998, Hippocampus; Bannerman et al., 2004, Neurosci Biobehav Rev; Bast, 2007, Rev Neurosci; Bast, 2011, Curr Opin Neurobiol; Fanselow and Dong, 2010, Neuron; Strange et al., 2014, Nature Rev Neurosci). Regarding the second statement, the authors may consider being more specific, as the inconsistencies demonstrated seem to relate mainly to the hippocampal representation of value information, instead of functional differentiation along the dorso-ventral hippocampal axis in general.

We agree with the reviewer that the abovementioned statements need further clarification. The manuscript is now revised as follows:

“Surprisingly, beyond the recognition of anatomical divisions, the available literature on the functional differentiation of subregions along the dorsoventral axis of the hippocampus, particularly in the context of value representation, is somewhat inconsistent.” (p.4)

Lines 92 to 93: 'Thus, it has been thought that the dHP is more specialized for precise spatial representation than the iHP and vHP.'I think 'fine-grained' may be the more appropriate term here. Also, check throughout the manuscript when referring to the differences of spatial representations along the hippocampal dorso-ventral axis.

Thank you for the insightful suggestion. We changed the term to ‘fine-grained’ throughout the manuscript, as follows:

“Thus, it has been thought that the dHP is more specialized for fine-grained spatial representation than the iHP and vHP.” (p.4)

“Consequently, the fine-grained spatial map present in the dHP…” (p.20)

Line 217: well-'trained' rats?

We initially used the term ‘well-learned’ to focus on the effect of learning, not training. Please note that the rats were already adapted to moving freely in the VR environment during the Shaping sessions, but the immediate counterclockwise body alignment only appeared after they acquired the reward locations for the main task. Nonetheless, we agree that the term might cause confusion, so we revised the manuscript as the reviewer suggested, as follows:

“This implies that well-trained rats aligned their bodies more efficiently…” (p.8)

Lines 309 to 311: 'Taken together, these results indicate that iHP inactivation severely damages normal goal-directed navigational patterns in our place preference task.'Consider to mention that dHP inactivation also causes impairments, albeit weaker ones.

We thank the reviewer for the suggestion. We revised the manuscript by mentioning dHP inactivation as follows:

“Taken together, these results indicate that iHP inactivation more severely damages normal goal-directed navigational patterns than dHP inactivation in our place-preference task.” (p.11-12)

Lines 550 to 552: 'The involvement of the iHP in spatial value association has been reported in several studies. For example, Bast and colleagues reported that rapid place learning is disrupted by removing the iHP and vHP, even when the dHP remains undamaged (Bast et al., 2009).'Bast et al. (2009) did not directly show the role of iHP in 'spatial value associations'. They suggested that the importance of iHP for behavioral performance based on rapid, one-trial, place learning may reflect neuroanatomical features of the intermediate region, especially the combination of afferents that could convey the required fine-grained visuo-spatial information with relevant afferent and efferent connections that may be important to translate hippocampal place memory into appropriate behavioral performance (this may include afferents conveying value information). More recent theoretical and empirical research suggests that projections to the (ventral) striatum may be relevant (see Tessereau et al., 2021, BNA and Bauer et al., 2021, BNA).

We appreciate the reviewer for this insightful comment. We agree with the reviewer that Bast et al. (2009) did not directly mention spatial value association; however, learning a new platform location needs an update of value information in the spatial environment. Therefore, we thought the study, though indirectly, suggested how the iHP contributes to spatial value associations. Nonetheless, to avoid confusion, we revised the manuscript, as follows:

“The involvement of the iHP in spatial value association has been reported or implicated in several studies” (p.20)

(2) Figures and legends:Figure 2B: What do the numbers after novice and expert indicate?

The numbers indicate the rat ID, followed by the session number. We added the details to the Figure legend, as follows:

“The numbers after ‘Novice’ and ‘Expert’ indicate the rat and session number of the example.” (p.34)

Figure 2C: Please indicate units of the travel distance and latency measurements.

The units are now described in the Figure legends, as follows:

“Mean travel distance in meters and latency in seconds are shown below the VR arena trajectory.” (p.34)

Figure 3Aii: Here and in other figures - do the vector lengths have a unit (degree?)?

No, the mean vector length is an averaged value of the resultant vectors, thus having no specific unit.

Figure 5A: Please explain what the numbers on top of the individual sample trajectories indicate.

The numbers are IDs for rats, sessions, and trials of specific examples. We added the explanation to the Figure legends, as follows:

“Numbers above each trajectory indicate the identification numbers for rat, session, and trial.” (p.35)

(3) Additional comments on some methodological details:a. Why was the non-parametric Wilcoxon signed-rank test used for the planned comparison between intermediate and dorsal hippocampal PBS infusions, whereas parametric ANOVA and post-hoc comparisons were used for other analyses? This probably doesn't make a big difference for the interpretation of the present data (as a parametric pairwise comparison would also not have revealed any significant difference between intermediate and dorsal hippocampal PBS infusions), but it would nevertheless be good to clarify the rationale for this.

We used the non-parametric statistics since our sample size was rather small (n = 8) to use the parametric statistics, although we used the parametric ANOVA for some of the results because it is the most commonly known and widely used statistical test in such comparisons. However, we also checked the statistics with the alternatives (i.e., non-parametric Wilcoxon signed-rank test to parametric paired t-test and parametric One-way RM ANOVA with Bonferroni post hoc test to non-parametric Friedman’s test with Dunn’s post hoc test), and the statistical significance did not change with any of the tests. We now added the explanation in the manuscript, as follows:

“Although most of our statistics were based on the non-parametric tests for the relatively small sample size (n = 8), we used the parametric RM ANOVA for comparing three groups (i.e., PBS, dMUS, and iMUS) because it is the most commonly known and widely used statistical test in such comparison. However, we also performed statistical tests with the alternatives for reference, and the statistical significances were not changed with any of the results.” (p.26)

b. Single housing of rats:Why was this chosen? Based on my experience, this is not necessary for studies involving cannula implants and food restriction. Group housing is generally considered to improve the welfare of rats.

We chose single housing of rats because our training paradigm required precise restrictions on the food consumption of individual rats, which could be difficult in group housing.

c. Anesthesia:Why was pentobarbital used, alongside isoflurane, to anesthetize rats for surgery (line 663)? The use of gaseous anesthesia alone offers very good control of anesthesia and reduces the risk of death from anesthesia compared to the use of pentobarbital.Why was anesthesia used for the drug infusions (line 674)? If rats are well-habituated to handling by the experimenter, manual restraint is sufficient for intra-cerebral infusions. Therefore, anesthesia could be omitted, reducing the risk of adverse effects on the experimental rats.I do not think that points b. and c. are relevant for the interpretation of the present findings, but the authors may consider these points for future studies to improve further the welfare of the experimental rats.

We appreciate the reviewer’s careful suggestions. For both the use of pentobarbital during surgery and anesthesia for the drug infusion, we chose to do so to avoid any risk of rats being awake and becoming anxious and to ensure safety during the procedures. They might not be necessary, but they were helpful for the experimenters to proceed with sufficient time to maintain precision. Nonetheless, we agree with the reviewer’s concern, which was the reason why we monitored the rats’ behavior for 20 minutes in the cage after drug infusion to minimize any potential influence on the task performance. We updated the relevant details in the Methods section, as follows:

“The rat was kept in a clean cage to recover from anesthesia completely and monitored for side effects for 20 minutes, then was moved to the VR apparatus for behavioral testing.” (p.25)